# Text-space Graph Foundation Models: Comprehensive Benchmarks and New Insights

**Zhikai Chen[1], Haitao Mao[1], Jingzhe Liu[1], Yu Song[1], Bingheng Li[1],**
**Wei Jin[2], Bahare Fatemi[3], Anton Tsitsulin[3], Bryan Perozzi[3],**
**Hui Liu[1], Jiliang Tang[1]**
[1]Michigan State University, [2]Emory University, [3]Google Research

## Abstract

Given the ubiquity of graph data and its applications in diverse domains, building a Graph Foundation Model (GFM) that can work well across different graphs and tasks with a unified backbone has recently garnered significant interests. A major obstacle to achieving this goal stems from the fact that graphs from different domains often exhibit diverse node features. Inspired by multi-modal models that align different modalities with natural language, the text has recently been adopted to provide a unified feature space for diverse graphs. Despite the great potential of these text-space GFMs, current research in this field is hampered by two problems. First, the absence of a comprehensive benchmark with unified problem settings hinders a clear understanding of the comparative effectiveness and practical value of different text-space GFMs. Second, there is a lack of sufficient datasets to thoroughly explore the methods' full potential and verify their effectiveness across diverse settings. To address these issues, we conduct a comprehensive benchmark providing novel text-space datasets and comprehensive evaluation under unified problem settings. Empirical results provide new insights and inspire future research directions. Our code and data are publicly available from `https://github.com/CurryTang/TSGFM`.

## 1 Introduction

Foundation Models (FMs) [1] have achieved remarkable success in various domains like computer vision [2, 3] and natural language processing [4, 5]. Through large-scale pre-training on diverse data [6, 7], FMs exhibit several intriguing properties compared to task-specific models trained in an end-to-end manner. First, one model can serve diverse tasks with better effectiveness [7, 8], and second, they present emergent capabilities such as in-context learning [9] and reasoning [10].

Nonetheless, the common practice in today's graph machine learning remains training task-specific models from scratch on each individual dataset [11]. Despite the success of graph models in diverse domains such as social networks [12, 13], e-commerce [14, 15, 16], and biology [17], most graph models still necessitate tailored data engineering and specific design for each dataset, which makes it hard to scale up due to limited data available for a single dataset [18].

Feature heterogeneity is the key obstacle for extending graph machine learning to training across data and tasks. [11]. Specifically, it refers to the fact that different graphs present different feature dimensions, where the corresponding dimension may have entirely different semantic meanings. Such a problem makes it impossible to train a GFM. To mitigate this problem, [19, 20] propose transforming different kinds of node attributes into texts and then using a large language model (LLM) to generate embeddings, which provides a unified feature space. This feature space offers two advantages: (1) it can mitigate the feature heterogeneity by mapping diverse node features into the same textual space, and (2) thanks to the rich latent knowledge in LLMs, the generated high-quality

text features may improve model performance [21, 22]. Leveraging these high-quality text features, we can unify different graphs in a manner akin to how multi-modal models unify modalities through text [2, 23, 24], thus giving rise to text-space GFMs [19, 20, 25]. Text-space GFMs can generalize to diverse graphs [19] and show preliminary success. Such a unified feature space also gives new potential for previous graph machine learning methods such as graph self-supervised learning [26] towards building GFM, which applies to various graphs and domains with a unified backbone.

Despite the considerable potential of text-space GFMs, a comprehensive understanding of their applicability and effectiveness across different application scenarios remains elusive.

a) First, most existing work is evaluated on a small number of datasets, primarily focusing on citation datasets, which makes the observations less representative and fails to reflect the full potential of GFMs.
b) Second, each work adopts its own GFM problem setting and proposes diverse GFM frameworks, which makes it hard to understand the effectiveness of different methods and hinders the development of a landscape of the whole field.
c) Third, existing work merely evaluates proposed methods, while understanding text-space GFMs' effectiveness remains elusive.

**Contributions.** To demystify the design spaces of text-space GFMs and inspire future research directions, we introduce a benchmark designed to illuminate the capabilities and limitations of existing text-space GFMs. Our contributions are multi-folded:

1. **Text-Space Datasets:** Recognizing the scarcity of existing text-space datasets and evaluation based on text space, we curate and preprocess over 20 datasets spanning academic, E-commerce, biology, and other miscellaneous domains.
2. **Comprehensive Evaluation on Diverse Use Cases:** Leveraging data from various tasks, we define four applicable GFM paradigms. We first evaluate different GFM building blocks under each setting and then adopt these building blocks as anchor models to investigate the overall effectiveness of text-space GFMs. Our benchmark provides a more comprehensive GFM setting than existing works.
3. **Novel Insights:** Our empirical results allow us to derive novel insights, and the most crucial ones are as follows: *Although LLMs offer a feature space with promising initial performance, there still exists gaps across different datasets. The positive transfer observed in text-space GFMs relies on transferable structural patterns and is only effective when combined with appropriate inductive biases designed for downstream tasks.*

## 2   Preliminaries

In this section, we introduce the background of our benchmark. First, we present the traditional graph machine learning (GML) pipeline, where a task-specific model is trained from scratch. Then, we delineate the general paradigms of text-space GFMs.

### 2.1   Problem Setting

**Traditional GML.** The standard GML setting involves training a *single model* for each task. Given dataset $P$ and downstream task $D$, a specific model $\mathcal{M}_t$ is trained on $P$ to address $D$. Such a pipeline necessitates specific data engineering and model deployment for each task.

**Graph Foundation Models.** GFMs extend the traditional GML setting across different datasets and tasks. Despite the more diverse settings, most GFMs follow a unified paradigm: transferring the knowledge from **training tasks** to tackle **downstream tasks** with a unified model backbone. Given a collection of training datasets $\mathcal{P} = \{P_1, P_2, \cdots, P_n\}$, where each dataset $P_i$ may encompass multiple training tasks $\mathcal{T}_i = \{T_{i1}, T_{i2}, \cdots, T_{ik_i}\}$, a GFM $\mathcal{M}_\theta$ is trained on the union of all training tasks $\mathcal{T} = \bigcup_{i=1}^{n} \mathcal{T}_i$ using a shared representation encoder $\mathcal{E}_\theta$ and optional task-specific heads $\mathcal{H} = \{H_{11}, H_{12}, \ldots, H_{nk_n}\}$ where $k_n$ represents the number of tasks for $n$-th dataset. The trained model $\mathcal{M}_\theta$ can then be adapted to tackle downstream datasets $\mathcal{D} = \{D_1, D_2, \cdots, D_m\}$, each with its own set of downstream tasks $\mathcal{S}_j = \{S_{j1}, S_{j2}, \cdots, S_{jl_j}\}$ where $l_j$ represents the number of tasks for $j$-th dataset. The adaption requires a unified architecture, which means either the entire model's parameters are shared or the encoder is shared with only tunable task-specific heads.

**Categorizing GFMs.** In this work, we draw inspiration from the GFM literature [19, 20, 11] to focus on the fine-grained categorization of GFMs in different use cases. We decompose these use cases using a framework of scenarios and tasks. A **scenario** describes the relationship between training and downstream tasks. In this work, we consider two scenarios: *co-training* and *pre-training*: **co-training** specifies the co-trained model to be applied to the same set of datasets, which means $\mathcal{P} = \mathcal{D}$. On the other hand, **pre-training** considers the case when pre-trained models are applied to novel datasets unseen in the training stage, which means $\mathcal{P} \neq \mathcal{D}$. Next, besides categorizing based on train and test data relationships, we use the concept of **tasks** to consider the relationship between training and downstream tasks. In this work, we consider conventional GML tasks, including node classification (NC), link prediction (LP), and graph classification (GC). Referring to [11], we categorize existing GFMs into **task-specific** and **cross-tasks** models. Task-specific GFMs focus on transferring inside a specific task, which means $T_1 = \cdots = T_n = S_1 = \cdots = S_m$. Cross-task GFMs target a more challenging setting where the knowledge is transferred across diverse tasks, such as node and graph classifications, which assumes the existence of $T_i \neq S_j$.

Based on the tuple (**scenarios**, **tasks**), we come up with 4 fine-grained GFM paradigms as shown in Figure 1. We then showcase the practical value of the proposed GFM paradigms.

**Practical value of the GFM paradigm.** GFMs have two primary strengths that we seek to leverage. First, their **efficiency**. GFMs aim to solve multiple tasks with one model, increasing developer velocity while decreasing maintenance complexity. Sharing a common model across tasks should allow for additional optimizations that would not be cost-effective in a one-model-per-task setting. In the **pre-training** scenario, GFMs with shared architecture can effectively adapt to a low-resource

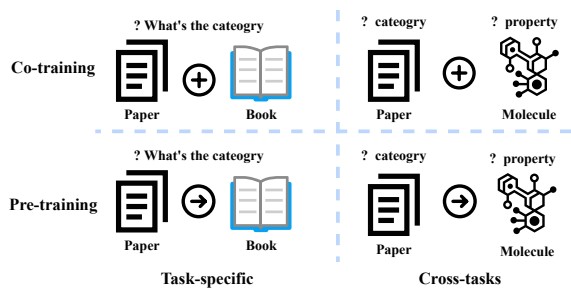

Figure 1: We come up with four paradigms: (Co-training, task-specific), (Co-training, cross-tasks), (Pre-training, task-specific), (Pre-training, cross-tasks)

downstream task without tuning parameters. Second, their **effectiveness**. GFMs have more model capacity and available training data than single-purpose models. Recent results show that increasing the amount of available training data can lead to better performance [18]. Especially in the **co-training** scenario, GFMs present the potential to improve performance by scaling across datasets and tasks.

## 2.2 Text-space GFM Building Blocks

When introducing general GFM paradigms, we emphasize using a unified model architecture to transfer, which requires a shared feature space across different datasets. To achieve a unified feature space, text-space GFMs adopt LLMs as the feature encoders, based on which various techniques have been proposed to learn transferable knowledge across different datasets and tasks, including graph SSL, graph prompts, and LLM with graph projectors [27].

**Text space as the unified feature space.** Text-space GFMs adopt LLMs as encoders to project node attributes into a unified feature space. However, this requires that the original attributes can be represented as texts. For non-text attributes like ones for molecules, text-space GFMs may adopt multi-modal models [19] like text-chemistry models [24] to transform the original attributes into texts. As a result, text-space GFMs can process a wide variety of datasets. We empirically evaluate the performance loss brought by the text-space transformation in Appendix B. Specifically, the unified text space provides us an opportunity to study the scaling capability [18, 28] of traditional GNN and emerging GFM models across different graphs, which extends the scope of previous works [18].

**Learning transferrable knowledge across graphs.** Building upon the unified feature space, various techniques have been proposed to learn transferrable knowledge across different graphs and tasks.

1. *Graph SSL* [26] employs a unified self-supervised learning task in the training stage, assuming that this task can learn general representations benefiting different downstream tasks.
2. *Foundational graph prompt* [19, 20, 29] transforms diverse tasks into a unified format. As a motivating example, [19] first unifies tasks at different levels by viewing node classification as the

ego-graph classification and link prediction as the classification of the node pair-induced subgraph. Then, it inserts tasks' labels as augmented nodes into the subgraph used for prediction, which converts multi-label classification into multiple binary classification problems, thereby unifying all classification tasks. Graph prompts mainly focus on unifying the formulation of tasks, and they still rely on the inductive bias of the model backbone to transfer across different graphs.

3. *LLM with graph projectors* [25, 30, 31, 32] leverages the inherent multi-task capability of LLMs. It is equipped with a projector from graph to text space [23], enabling natural language to describe different graph tasks and thus achieving a unified task formulation.

## 3  Text-space Dataset

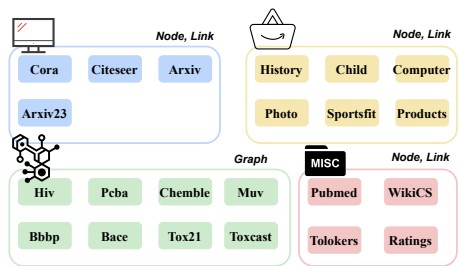

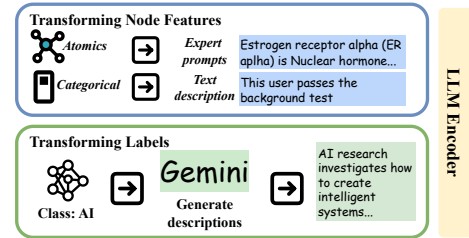

Figure 2: Our proposed text-space dataset covering 20+ datasets coming from diverse domains.

Figure 3: Transforming attributes and labels into text space.

To facilitate a comprehensive evaluation of diverse paradigms (Section 2.1), we introduce over 20 text-space datasets as shown in Figure 2. These are derived from [22, 24, 19, 33, 34, 35], with attributes transformed into texts through pre-processing of raw files or the generation of expert descriptions following [24, 19] as shown in Figure 3.

**Dataset pre-processing.** We generally follow [19, 24] to conduct attribute pre-processing. For example, given an E-commerce dataset, we set the node attribute to **"Feature node. Product Title: <product_title>"** (<product_title> is the text-space attribute we collect), edge attribute to **"Feature edge. These two items are frequently co-purchased or co-viewed"**. For molecular graphs, we adopt the text descriptions generated by [24]. Label description is generated by LLMs, with examples like **"The "Case-Based" category refers to research papers focusing on case-based reasoning (CBR) in the field of artificial intelligence..."**. We utilize Gemini [36] to generate text descriptions for labels. We double-check the outputs of LLMs to ensure the quality of their generation. Then, we encode these attributes into corresponding feature, edge, and label embedding using a text encoder model [37].

These datasets encompass a variety of tasks, including node classification, link prediction, and graph classification. Leveraging MMD [38] as a similarity metric and considering the source of datasets, we categorize them into domains. This yields 4 datasets from the CS citation domain, 6 from e-commerce, and 8 from the molecular domain, with the remaining classified as other domains due to divergence from established ones. This categorization allows us to investigate two key questions: (1) For those in-domain datasets with similar features, can text-space GFM fully address feature heterogeneity and achieve positive transfer? (2) Does increasing the volume of training data, both within and across domains, improve GFM performance, thereby demonstrating neural scaling properties [28]? We adhere to the original splits [22, 34, 19, 33] to simulate varying dataset sizes in real-world applications. Notably, our contribution lies in the breadth and diversity of text-space datasets across domains, exceeding the scope of prior works [20, 19, 25]. Detailed dataset descriptions are provided in Appendix D.

## 4  Empirical Studies

In this section, we present the empirical studies of text-space GFMs. Based on the problem setting in Section 2, we conduct research from the following two dimensions: (1) In each of the 4 paradigms, we comprehensively evaluate different building blocks of GFMs. (2) Based on the experimental results, the selected datasets and models can be viewed as anchors to reflect the overall effectiveness

of text space GFMs in this paradigm. The following subsections will be structured as follows: we first introduce the general experiment configurations and then present the empirical results.

## 4.1 Experiment Configurations

We first present the selected GFM building blocks and evaluation settings.

**Models.** Following Section 2.2, we adopt the following models for each GFM building block:

1. For Graph SSL, we adopt representative methods including DGI [39], GCC [40], BGRL [41], and also GraphMAE [42]. These methods cover different paradigms such as contrastive learning-based SSL, augmentation-free SSL, and feature reconstruction-based SSL [26]. To train these models across different graphs, we adopt GraphSAINT [43] to extract mini-batches with size 1024.
2. For foundational graph prompt models, we adopt two representative methods, OFA [19] and Prodigy [20], specifically designed for GFM training. The original OFA introduces weights to balance different datasets, which are not proportional to the size of the dataset and require extensive tuning, making them impractical in real-world scenarios. We set all weights to 1 to examine the model's preference across different data and tasks.
3. For LLM with graph projectors, we adopt LLaGA [25] considering its effectiveness and simplicity. We adopt Mistral-7B [44] as the LLM backbone.
4. As link prediction can transfer across different graphs with a unified formulation, we consider link prediction-specific models like BUDDY [45] and SEAL [46] for link prediction.

For the LLM encoder, we adopt Sentence-BERT [37] since it can achieve good performance with low computational cost [22]. We discuss how other LLM encoders affect the results in Appendix G.2.

**Evaluation settings.** We use the performance on downstream tasks to evaluate different GFMs. Specifically, for node-level tasks, we use accuracy as the metric. We use the corresponding metrics used in [34] for graph-level tasks. Notably, we use the hit rate as the metric for link-level tasks. [19, 20, 25] use AUC and accuracy to evaluate link prediction, which has been shown ineffective in differentiating different baselines [47]. For hyper-parameter tuning, different hyper-parameters lead to varying model preferences across datasets. Therefore, we utilize the average validation performance of different datasets to select the optimal model. We present the comprehensive experimental settings and model-specific hyper-parameter searching range in Appendix E.

The following subsections present the empirical evaluation results following four paradigms in Section 2.1. We first present the specific experimental settings and the empirical results. At the end of each paradigm, we highlight the core observations. In this paper, we focus our investigation on the co-training setting for two main reasons: **First**, co-training is a natural extension of the existing end-to-end learning paradigm on graphs, allowing us to leverage existing principles [11] for understanding and making it an actionable next step. **Second**, through effective adaptation techniques [48], co-trained models also have the potential to be applied to the pre-training setting.

## 4.2 Case 1: Co-training over the same task

We start from the paradigm (**Co-training**, **Task-specific**). This work mainly focuses on three tasks: node classification, link prediction, and graph classification.

### 4.2.1 Co-training for Node Classification

**Experiment Settings.** For co-training over node-level datasets, we adopt graphs from the CS Citation domain, E-commerce domain, Pubmed, and WikiCS from other domains. For baseline models, we consider all baselines introduced in 4.1 except Prodigy and link prediction-specific methods, which are not applicable. We evaluate models under the following three settings: (1) the model is trained on a specific downstream task from scratch; (2) the model is co-trained on graphs from the same domain; and (3) the model is co-trained on overall available datasets.

**Results.** We summarize the performance of each model on individual datasets after co-training in Table 1. As the performance of BGRL and GCC are significantly lower than other methods, we omit them from the table for visualization clarity. Our results indicate that various GFM methods, regardless of in-domain or cross-domain co-training, still underperform compared to task-specific GCN baselines. Notably, LLaGA and OFA, based on supervised learning, exhibit better overall performance and surpass GCN baselines in the E-commerce domain. Meanwhile, we find that different methods exhibit different characteristics during in-domain and cross-domain co-training as

Table 1: Performance of node-level co-training. ST refers to "training on a single graph from scratch". ID refers to "co-training on graphs coming from the same domain". CD refers to "co-training across all graphs". "Cit-Avg" records the average performance of citation datasets. "Ecom-Avg" records the average performance of E-commerce datasets. "Avg" records the average performance of all datasets. green and yellow represent the domain of data. Underline represents the case where co-training benefits compared to training from scratch.

| Methods | Setting | Cora | CiteSeer | Arxiv | Arxiv-2023 | History | Child | Photo | Computers | Sports | Products | Cit-Avg | Ecom-Avg | Avg |
|---|---|---|---|---|---|---|---|---|---|---|---|---|---|---|
| GCN | ST | 82.20 | 75.29 | 73.10 | 74.98 | 85.25 | 56.62 | 82.42 | 87.43 | 89.37 | 88.00 | 76.39 | 81.52 | 79.47 |
| OFA | ST | 79.41 | 81.35 | 73.85 | 73.75 | 83.33 | 53.77 | 84.46 | 86.48 | 92.50 | 87.35 | 77.09 | 81.32 | 79.63 |
|  | ID | 70.74 | 81.66 | 72.68 | 74.07 | 83.30 | 56.22 | 85.05 | 87.83 | 92.29 | 86.91 | 74.79 | 81.93 | 79.08 |
|  | CD | 72.63 | 70.19 | 72.72 | 74.13 | 83.88 | 56.89 | 84.95 | 87.65 | 92.35 | 86.96 | 72.42 | 82.11 | 78.24 |
| GraphMAE | ST | 81.00 | 74.36 | 71.67 | 74.40 | 83.07 | 51.79 | 83.27 | 83.54 | 88.49 | 85.90 | 75.36 | 79.34 | 77.75 |
|  | ID | 78.09 | 68.80 | 72.80 | 73.30 | 83.82 | 51.38 | 83.47 | 83.82 | 88.47 | 85.90 | 73.25 | 79.48 | 76.99 |
|  | CD | 80.27 | 70.65 | 72.43 | 71.02 | 83.95 | 51.38 | 83.00 | 83.39 | 88.38 | 85.88 | 73.59 | 79.33 | 77.04 |
| DGI | ST | 81.80 | 72.95 | 70.36 | 72.47 | 82.93 | 48.34 | 83.38 | 80.86 | 86.28 | 84.14 | 74.40 | 77.66 | 76.35 |
|  | ID | 80.17 | 67.18 | 71.39 | 72.88 | 83.11 | 49.45 | 81.78 | 82.90 | 86.77 | 85.47 | 72.91 | 78.25 | 76.11 |
|  | CD | 81.50 | 73.14 | 71.84 | 72.44 | 83.24 | 49.64 | 83.25 | 82.68 | 86.67 | 85.21 | 74.73 | 78.45 | 76.96 |
| LLaGA | ST | 81.25 | 68.80 | 76.05 | 76.00 | 82.55 | 55.05 | 86.00 | 87.75 | 91.45 | 88.85 | 75.53 | 81.94 | 79.38 |
|  | ID | 79.10 | 68.25 | 76.20 | 75.80 | 83.30 | 54.45 | 85.40 | 87.00 | 91.40 | 89.00 | 74.84 | 81.76 | 78.99 |
|  | CD | 76.45 | 63.95 | 75.90 | 75.10 | 81.80 | 54.10 | 86.60 | 86.75 | 90.60 | 88.80 | 72.85 | 81.44 | 78.01 |

follows: (1) When co-training on the same domain, LLaGA tends to match the performance of training from scratch. Cross-domain co-training on a large number of datasets only negatively impacts LLaGA. A similar phenomenon can also be observed when LLMs with cross-modality projectors are applied to CV [49], which may be related to catastrophic forgetting. (2) We observe that hyperparameter tuning can improve the performance of model training from scratch. However, the optimal hyperparameters vary across datasets, contributing to the underperformance of the unified co-trained model. (3) Co-training can potentially benefit SSL methods. Specifically, DGI demonstrates the potential for performance improvement with increasing data scale. The key observations can be summarized as follows.

**Observation 1.** *Under the task-specific co-training for node classification, GFM methods present a performance gap compared to GCN training from scratch, while certain methods like DGI show potential to improve performance with data scale.*

**Further Probing.** To better understand the ineffectiveness of node-level co-training, we further investigate the design of OneForAll, the model with the best performance. We consider two surrogate models to disentangle the influence of node features and graph structures: (1) replacing OneForAll's backbone with MLP to eliminate the graph structures and (2) replacing OneForAll's GCN-based backbone with SGC [50]-like fixed feature preprocessing. As shown in Table 2, three different sets of data result in three distinct outcomes. For the citation dataset, we observe a decrease in MLP and GCN's performance after co-training, indicating that even without the influence of structure, features in this dataset still lead to negative transfer. For e-commerce datasets, there is no negative transfer for both MLP and GCN. Using SGC to replace the GCN backbone yields better results in all three cases. The primary reasons why we don't observe benefits in node-level co-training are: (1) Stacking more data doesn't exhibit a scaling behavior if we ignore graph structure; (2) When considering graph structure, GCN with learnable aggregation as the backbone does not perform better than SGC [50] with fixed aggregation, indicating that stacking more data does not lead to learning a better aggregation function. Since there is no improvement in either feature or structure aspects, co-training shows no benefits.

Table 2: Average performance is recorded in the table. ST means the model is trained from scratch on a single graph. CT means co-training across different graphs. GCN-* represents the original OneForAll model, while SGC-* represents the variants replacing the original GCN backbone with SGC backbone.

| Co-train: CS + Pubmed (citation) | | | | | Co-train: E-commerce | | | | | Co-train:All | |
|---|---|---|---|---|---|---|---|---|---|---|---|
| GCN-ST | GCN-CT | MLP-ST | MLP-CT | SGC-CT | GCN-ST | GCN-CT | MLP-ST | MLP-CT | SGC-CT | GCN-CT | SGC-CT |
| 75.20 | 69.97 | 71.62 | 68.33 | 72.51 | 81.32 | 81.93 | 71.89 | 71.83 | 82.9 | 76.31 | 80.01 |

### 4.2.2  Co-training for Link Prediction

**Experiment Settings.** Following the setting of node-level co-training, we adopt graphs from the CS Citation and E-commerce domains for co-training over link prediction tasks to consider the impact of data quantity and domain. For baseline models, we select GraphMAE as a representative for graph SSL, considering its superior performance compared to other SSL methods. We also include OFA and

LLaGA, which apply to this paradigm. Since different datasets share the same task formulation, we also adopt end-to-end GCN, SEAL, and BUDDY for co-training (which can be seen as task-specific GFMs). Considering the efficiency of existing GFM pipelines, we first evaluate all methods under three small-scale datasets: Cora, CiteSeer, and Pubmed. Then, we extend scalable methods to co-train on larger graphs.

**Results.** The results of different methods on small-scale datasets are presented in Figure 4 and Table 3. GFM methods demonstrate no advantages compared to link prediction-specific models regarding efficiency and effectiveness. Comparing different GFMs, LLaGA achieves the best performance, but there is still a significant gap compared to link prediction-specific models like BUDDY. One reason for this phenomenon is that link prediction requires modeling the task-specific inductive bias revolving on the pairwise structural patterns [11], while these patterns are largely ignored by the GFM with a unified architecture across tasks. This also suggests that designing a task-specific GFM for link prediction could be a promising direction. Meanwhile, we notice a considerable gap between SEAL and BUDDY, which suggests that properly incorporating structural embeddings is crucial for achieving optimal performance.

We further extend the scalable model BUDDY and GCN to larger graphs for co-training, with the results presented in Table 4. GraphMAE is omitted due to its poor performance on E-commerce datasets. The experimental results demonstrate that co-training significantly benefits models like BUDDY, which leverages suitable structural features. As a comparison, GCN achieves much worse performance and co-training shows no clear benefits compared to training on a single graph from scratch. Experimental results indicate that to achieve positive transferring from co-training on link prediction tasks, models need to incorporate proper inductive bias through structural embeddings. We note that even though SEAL's performance is not satisfying in Table 3, co-training still enhances performance across all downstream datasets. We summarize the aforementioned discussions with the following key observations.

**Observation 2.** *GFM methods show no advantages over link prediction-specific models and struggle to scale to large graphs. Link prediction-specific models like BUDDY show great potential to benefit from co-training, highlighting the importance of proper structural embeddings. This also suggests that designing a task-specific GFM for link prediction could be a promising direction.*

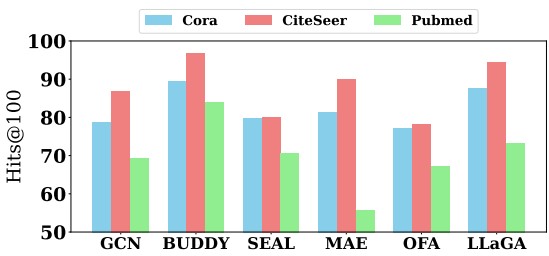

Figure 4: Comparison of different GFM and link prediction-specific models **co-trained** on three small-scale graphs. Hits@100 is adopted as the metric.

Table 3: Comparison of different models' average performance trained on a single graph or co-trained on Cora, CiteSeer, Pubmed. We omit the feature preprocessing time for BUDDY and SEAL.

|         | Single-task | Co-train | Training time (s) |
|---------|-------------|----------|-------------------|
| GCN     | 79.39       | 78.25    | 49                |
| SEAL    | 70.40       | 76.78    | 3492              |
| MAE     | 80.48       | 74.80    | 32                |
| OFA     | 75.11       | 74.05    | 5431              |
| LLaGA   | 81.32       | 85.03    | 7209              |
| BUDDY   | 90.41       | 90.05    | 148               |

Table 4: Co-training BUDDY and GCN for link prediction at scale. -S means the model is trained on a single downstream task. -D means the model is trained on data coming from similar domains (shown in green and yellow). We use underline to emphasize the case where co-training benefits compared to training from scratch.

|         | Cora  | Citeseer | Arxiv | Arxiv23 | History | Child | Photo | Computers | Sports | Products | Average |
|---------|-------|----------|-------|---------|---------|-------|-------|-----------|--------|----------|---------|
| **BUDDY-S** | 91.37 | 96.57 | 86.91 | 90.00 | 75.57 | 58.22 | 73.97 | 74.44 | 77.00 | 30.78 | 75.48 |
| **BUDDY-D** | 88.72 | 97.28 | 67.04 | 89.22 | 91.17 | 86.25 | 90.42 | 88.72 | 93.6 | 69.34 | 86.18 |
| **GCN-S** | 83.12 | 88.91 | 25.13 | 75.77 | 45.01 | 16.41 | 36.02 | 24.23 | 24.65 | 13.46 | 43.27 |
| **GCN-D** | 73.46 | 84.73 | 47.61 | 81.75 | 33.07 | 7.95 | 38.55 | 41.69 | 7.95 | 6.18 | 42.29 |

### 4.2.3 Co-training for Graph Classification

**Experiment Settings.** We adopt all available text-space datasets from the molecular domain for graph-level co-training. We adopt OFA and GraphCL as baseline models. The former is implemented

based on foundational graph prompts, while the latter is based on self-supervised learning. We compare models co-trained on different datasets with models trained from scratch on single datasets. We consider models trained on original atomic and text features for the latter. It's important to note that **our dataset primarily focuses on tasks related to molecular property prediction**, and the results in other domains warrant further investigation.

Table 5: Performance of graph-level co-training. Underline represents the best results on each dataset.

| | PCBA | HIV | TOX21 | BACE | BBBP | MUV | TOXCAST |
|---|---|---|---|---|---|---|---|
| Single (Atom) | 0.202 | 75.49 | 74.6 | 72.4 | 65.7 | 70.7 | 61.5 |
| Single (Text) | 0.174 | 74.2 | 74.49 | 72.25 | 67.71 | 64.88 | 60.24 |
| GraphCL (co-train) | 0.203 | 70.33 | 73.74 | 64.42 | 63.09 | 73.62 | 60.77 |
| GraphCL (pre-train) | N/A | 78.47 | 73.87 | 75.38 | 69.68 | 69.8 | 62.4 |
| OFA | 0.236 | 75.24 | 82.5 | 77.32 | 69.97 | 70.39 | 68.39 |

**Results.** As shown in Table 5, co-training brings clear benefits for graph-level tasks. After unifying the feature and task formulation, models surpass the single dataset counterpart on all datasets after co-training. We also notice that in the single dataset case, there is a performance gap between the model using LLM features and the model using original features. This indicates there's still some performance loss by transforming original attributes into text space. After co-training, the gap is eliminated, and the co-trained model based on text features performs better. If we compare OneForAll to GraphCL under the co-training setting, we find that OneForAll achieves better performance, probably due to the unification of label space. GraphCL, as a comparison, achieves good performance only when pre-trained on large-scale data like ZINC.

**Observation 3.** *Co-training in the text space brings clear performance gain compared to training from scratch for graph classification tasks on molecular datasets.*

### 4.3  Case 2: Co-training across tasks

We study the paradigm (**Co-training**, **Cross-task**) in this section. This setting is more challenging than task-specific co-training, requiring modeling shared principles across different tasks. We consider two settings: first, cross-task co-training happening on the same set of graphs, corresponding to node-level and link-level co-training, which we relegate to Appendix G.1.1. The second scenario is cross-task co-training across graphs, as shown below.

#### 4.3.1  Co-training across node classification, link prediction, and graph classification

Co-training over node classification and link prediction still focuses on the paradigm of cross-task co-training on the same graph. We then investigate node, link, and graph-level co-training across different tasks and different graphs.

**Experiment Settings.** We adopt all datasets from node classification co-training for node-level datasets (Section 4.2.1, three small-scale datasets for link-level datasets (Section 4.2.2), and all datasets from graph classification co-training (Section 4.2.3) for graph-level datasets. Detailed settings can be found in Appendix G.1.1.

**Results.** As shown in Table 6, we observe that node and graph co-training, link and graph co-training, or node-link and graph co-training, all significantly improve graph-level performance, but they do not provide benefits for node-level or link-level tasks. Notably, co-training with tasks like link prediction that do not present node-level annotations can also significantly benefit graph-level tasks. However, co-training does not improve and may even degrade node-level performance.

Table 6: Performance of cross-data cross-task co-training. Link-Graph means co-training over link-level and graph-level tasks. The remaining two columns follow the same naming convention. We separate PCBA from the other **graph datasets** due to the significant difference in the scale of results. Underline means cross-task co-training benefits compared to single-task co-training.

| Single task | | | | Link-Graph | | | Node-Graph | | | Link-Node-Graph | | |
|---|---|---|---|---|---|---|---|---|---|---|---|---|
| Link Avg | Node Avg | PCBA(G) | Graph Avg | Graph Avg | PCBA(G) | Link Avg | Graph Avg | PCBA(G) | Node Avg | Graph Avg | PCBA(G) | Node Avg |
| 74.05 | 78.24 | 0.233 | 72.24 | 75.04 | 0.265 | 74.04 | 75.86 | 0.282 | 76.43 | 75.48 | 0.279 | 76.06 |

**Observation 4.** *When co-training OFA on node classification, link prediction, and graph classification tasks across different datasets, the model's performance in graph classification will improve while its performance at link prediction and node classification may decline.*

The primary reason behind this phenomenon is that OFA tends to learn inductive biases that are more suitable for graph-level tasks. In that way, the model leverages structural information in node-

and link-level tasks to enhance graph-level performance. However, this emphasis on structure may introduce noise that can negatively impact performance on node-level tasks. We put the detailed discussion in Appendix G.1.2.

## 4.4 Case 3 & 4: Transferring from pre-training to downstream tasks

In this subsection, we consider the **pre-training** scenario, where the primary distinction from co-training lies in the absence of overlap between the pre-training and downstream datasets. Foundation models in other domains have demonstrated two potential capabilities in this setting: (1) the ability to enhance downstream task performance through a pre-train and fine-tune paradigm [4], and (2) the ability to learn in context [51] on downstream tasks.

**General Experiment Settings.** To assess the effectiveness of different GFMs, we adopt two evaluation protocols: in-context learning (zero-shot and few-shot) and fine-tuning. For in-context learning, we assume the downstream task has no labels (zero-shot) or only $k$ labels per class (fewshot, $k = 3$ in this section). For fine-tuning, we assume the same labeling rate as in co-training. Specifically, we utilize the three largest node-level datasets, Arxiv, Sportsfit, and Products, as the pre-training data. We then employ Cora, History, and Amazon ratings to evaluate node classification downstream task performance, PubMed for link prediction task performance, and HIV for graph classification task performance. For graph SSL, we use the level of the labels provided by the train data as the level for pre-training because the learned representation will conform to the corresponding inductive bias [52].

**Foundational graph prompts for transferring.** As shown in Section 2.2, the foundational graph prompt is widely adopted by GFMs under the pre-training setting. It aims to achieve better transferring effectiveness by narrowing the gap between pre-training and downstream task formats. To gain a deeper understanding of its effectiveness, we investigate various types of graph prompts, including GPPT [53], GraphAdapter [54] and GPrompt [29] in the original label space and Prodigy [20] and OneForAll [19] in the LLM embedding space.

**Roles of LLM for transferring.** To further study the effectiveness of LLMs for pre-training, we also consider LLM-based methods GraphLLM [22] and GraphText [55]. The former directly uses text attributes as input, while the latter constructs prompt input containing graph inductive bias through the clustering property of attributes. We also consider a model variant, "SimpleSBERT", that does not require pre-training. For zero-shot learning, it obtains node embeddings through feature propagation and selects the nearest label embedding in the feature space as the corresponding prediction. For few-shot and fine-tuning settings, it's equivalent to a normal GCN.

### 4.4.1 Case 3: Transferring across the same tasks.

Table 7: Performance of transferring across the same task. The **N/A** in the table indicates that the model is not applicable to this setting. The best performance is shown in **bold** text.

|  | Cora | | | History | | | Ratings | | |
|---|---|---|---|---|---|---|---|---|---|
|  | 0 shot | 3 shot | FT | 0 shot | 3 shot | FT | 0 shot | 3 shot | FT |
| GraphMAE | N/A | 72.49 | 81.8 | N/A | 39.15 | 83.68 | N/A | 31.68 | 41.06 |
| LLaGA | 18.25 | 60.7 | 80.45 | 22.05 | 36.45 | 82.55 | 23.15 | 23.45 | 28.2 |
| OFA | 30.42 | 52.49 | 74.27 | 22.98 | 39.36 | 83.53 | 21.72 | 29.08 | **51.44** |
| OFA-FS | 20.3 | 42.1 | N/A | 13.82 | 17.5 | N/A | 21.5 | 20.5 | N/A |
| Prodigy (MAG240M) | N/A | 23.4 | N/A | N/A | 12.71 | N/A | N/A | 20.16 | N/A |
| Prodigy | N/A | 40.59 | N/A | N/A | 19.47 | N/A | N/A | 20.84 | N/A |
| GraphText | N/A | 50.33 | N/A | N/A | 48.00 | N/A | N/A | **37.67** | N/A |
| GraphAdapter | 21.09 | 33.74 | 62.56 | 17.43 | 36.60 | 82.94 | 27.64 | 29.41 | 38.69 |
| GraphLLM | 67.33 | 68.00 | N/A | 34.67 | **56.67** | N/A | 24.33 | 34.33 | N/A |
| GPPT | N/A | 44.14 | 65.84 | N/A | 27.54 | 35.94 | N/A | 14.24 | 20.22 |
| Gprompt | N/A | 55.38 | 70.82 | N/A | 17.36 | 21.33 | N/A | 15.38 | 17.23 |
| Simple SBERT (no pretrain) | **67.41** | 68.42 | **82.2** | **59.25** | 51.25 | **85.3** | 27.39 | 20.95 | 48.46 |

**Experiment Settings.** We start from the case where transferring happens between the same task. Since the pre-training dataset contains node-level labels, we evaluate the node-classification task. Following the general settings in Section 4.4, we select baseline models GraphMAE, LLaGA, OFA, GraphText, GraphLLM, GPPT, GPrompt, and Prodigy applicable to the transferring settings. "SimpleSBERT" is also considered to evaluate the effectiveness of pre-training. For OFA, we consider the normal prompt version and one designed for few-shot inference. For Prodigy, we consider the version pre-trained on the same dataset as other baselines or the one trained on MAG240M as in the original paper.

**Results.** The results are shown in Table 7; we first notice that the "pre-training, fine-tuning" paradigm doesn't present clear benefits with only marginal gain on heterophilous dataset Amazon ratings, which may be explained by the fact that LLM embeddings are powerful enough to achieve good downstream task performance with only a small number of labels, rendering pre-training not that helpful for providing additional information.

Table 8: Performance of transferring across different tasks. We use Hits@100 for link prediction and AUC-ROC for graph classification.

| | Pubmed | | HIV | | |
|---|---|---|---|---|---|
| | 0 shot | FT | 0 shot | 3 shot | FT |
| GraphMAE | N/A | 36.83 | N/A | 56.68 | 65.39 |
| LLaGA | 14 | 74 | N/A | N/A | N/A |
| OFA | 0.49 | 68.7 | 49.73 | 50.41 | 75.35 |
| OFA-FS | 4.56 | N/A | 47.8 | 50.56 | N/A |
| Simple SBERT (no pretrain) | 49.28 | 66.14 | N/A | N/A | 74.2 |

One surprising phenomenon lies in the in-context learning setting, where we observe a large gap between OFA and our proposed "simple SBERT" model. We observe that simply adopting LLM embeddings with feature propagation can outperform almost all graph in-context learning on homophilous graphs, which indicates:

**Observation 5.** *Relying on textural space, pre-trained LLM embedding is the key to support in-context learning.*

Moreover, we notice that semantic label embeddings are critical to the transferability of graph prompts. Otherwise, they fail, like GPPT, Prodigy, and GPrompt.

### 4.4.2 Case 4: Transferring across tasks.

**Experiment Settings.** We then study the paradigm where transferring happens across different tasks. We select GraphMAE, LLaGA, OFA, and our proposed simple baselines in Section 4.4.1.

**Results.** As shown in Table 8, we observe that (1) existing GFMs present limited capability in cross-task in-context learning. (2) After fine-tuning, we observe positive transferring from node classification to graph classification, which is consistent with our observation in Table 6.

**Observation 6.** *We are still far away from cross-graph, cross-task in-context learning.*

## 5 Conclusion

This paper presents a novel benchmark designed for developing text-space GFMs, which comprise datasets, comprehensive evaluation under diverse settings, and novel insights. Our key findings can be summarized as follows: the effectiveness of text-space GFM is built on three conditions: (1) LLM embeddings provide a feature space mitigating severe negative transferring. (2) GFM models can extract transferrable patterns across different graphs. (3) GFM backbones present appropriate inductive biases designed for downstream tasks. Insights from our work can potentially inspire research in diverse areas, including E-commerce, social networks, and natural science. We present a thorough discussion on broader impacts in Appendix I.

## 6 Acknowledgements

Zhikai Chen, Haitao Mao, Jingzhe Liu, Yu Song, Bingheng Li, Hui Liu, and Jiliang Tang are supported by the National Science Foundation (NSF) under grant numbers CNS2321416, IIS2212032, IIS2212144, IOS2107215, DUE2234015, CNS2246050, DRL2405483 and IOS2035472, the Army Research Office (ARO) under grant number W911NF-21-1-0198, Amazon Faculty Award, JP Morgan Faculty Award, Meta, Microsoft and SNAP.

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

# Appendix

## Table of Contents

# A More backgrounds of our benchmark

## A.1 Components of our benchmark

As shown in 5, our benchmarks compose diverse datasets, implementation of GFM building blocks, and comprehensive evaluation.

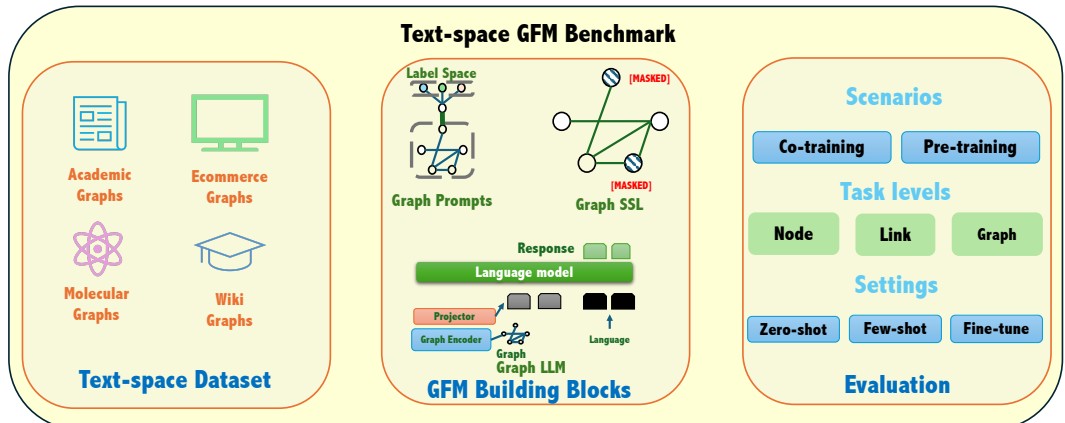

Figure 5: Our benchmark comprises three main components: (1) **Diverse text-space datasets**: Covering 23 text-space datasets from diverse domains; (2) **GFM building block:** Implementation of mainstream techniques to build GFMs; (3) **Comprehensive Evaluation:** We propose four use cases to evaluate the performance of GFMs thoroughly.

## A.2 Comparison between our benchmark and existing works

Table 9: Comparison between our benchmark and existing works: We present many more text-space datasets, based on which we consider comprehensive problem settings of GFM. We adopt graph SSL and link prediction-specific methods, which have often been overlooked in other works. We also employ reasonable experimental settings, such as comparing GFMs with GNNs using LLM embeddings and ensuring no test edge leakage in link prediction evaluation. Finally, we propose new understandings based on reliable experimental results.

| | Diverse Datasets | Comprehensive settings | Comprehensive Baselines | Comprehensive Evaluation | Understanding |
|---|---|---|---|---|---|
| GraphGPT [30] | ✗ | ✗ | ✗ | ✗ | ✗ |
| LLaGA [25] | ✗ | ✗ | ✗ | ✗ | ✗ |
| Prodigy [20] | ✗ | ✗ | ✗ | ✗ | ✗ |
| OneForAll [19] | ✔ | ✔ | ✗ | ✗ | ✗ |
| UniGraph [56] | ✗ | ✔ | ✔ | ✗ | ✗ |
| Ours | ✔ | ✔ | ✔ | ✔ | ✔ |

## A.3 Scope of the paper

Our paper investigates two usage scenarios for GFMs: co-training and transferring, and focuses on the co-training phase due to: (1) the efficiency issue of most current GFMs limits large-scale pre-training/transferring research, hindering rigorous conclusions. Co-training on downstream data and comparing it with model training from scratch can help us better compare the potential of text space models. (2) Achieving improved performance via co-training is a more realizable goal in the current stage. Transferring to new data involves problems like catastrophic forgetting [57] and distribution shift, which is more challenging.

# B An Empirical Investigation into Performance Degradation from Projecting into Text Space

In this section, we empirically evaluate the performance loss when transforming diverse kinds of attributes into text space. Specifically, we evaluate the following cases as shown in Table 10.

Table 10: Performance comparison between models trained on original attribute space and text space

|          | Original Attributes | Task                 | Metric   | Original Performance | Text-space Performance |
|----------|---------------------|----------------------|----------|----------------------|------------------------|
| **Arxiv**    | Word2Vec            | Node Classification  | Accuracy | 71.53                | 73.10                  |
| **HIV**      | Atomic Numbers      | Graph Classification | AUC-ROC  | 75.52                | 74.20                  |
| **Tolokers** | Categorical         | Node Classification  | Accuracy | 83.25                | 78.16                  |
| **Pubmed**   | TF-IDF              | Link Prediction      | Hits@100 | 53.05                | 66.13                  |

The experimental results demonstrate:

- For text attributes, LLMs can generate better-quality embeddings and empower tasks like node classification and link prediction.
- For non-text attributes like atomic numbers and categorical values, high-quality text prompts can achieve comparable performance.

## C    Comprehensive Related Works

### C.1    Graph Foundation Models (GFMs)

Despite the diverse definitions and scopes of existing GFMs [58, 59, 11], one core criterion defining a GFM is the capability to empower a series of graph-related tasks with a unified backbone. This unified backbone can either be trained from scratch, making it a graph-centric GFM [58]; or it can be adapted from an existing foundation model (mostly LLM), making it an LLM-induced GFM [27].

Graph-centric GFM's scope is mostly focused on traditional graph machine learning tasks, and its core philosophy is to unify diverse data and tasks to enlarge the training data, thereby empowering models' capability and also enabling "one model serves all". The core challenge lies in the diversity of graph structures and node features. Tackling diverse graph structures is a trending topic in today's graph machine learning, with models focused on proposing backbones [60] with more flexible inductive biases or enhancing existing backbones to tackle more diverse structures [61]. However, feature heterogeneity has been studied less, and there is currently no solution that can handle all different scenarios well. At the same time, feature heterogeneity is so critical that without a unified feature space, it's impossible to train a GFM. To tackle this issue, some approaches have either ignored feature information altogether, leading to significant performance drops on text-attributed graphs [62, 63], or constructed domain-specific feature spaces for knowledge graphs or molecules, limiting their generalizability [59, 64, 65]. In contrast, *Text-space* GFM, which transforms diverse attributes into texts and then adopts large language models (LLMs) as encoders, [20, 19, 25], provides a unified feature space that can generalize to a wide range of graphs [19] and demonstrate impressive performance. Despite the preliminary success, our understanding of text-space GFM is still pretty limited. For instance, the tasks on which they have better effectiveness and under what circumstances they can achieve positive transfer, these gaps in understanding motivate us to conduct this benchmark. Moreover, the text space also presents limitations. While most node features can be converted into text, sometimes, this can lead to significant performance loss. Additionally, how to model the interaction between graph structure and LLM features remains an open research question. Another problem lies in the capability of the backbone. Despite the wide applicability of message-passing NN in diverse applications, they still present fundamental capacity limitations, which are addressed by Transformer architectures [66]. [67, 68, 69] studies the transferability of GNNs specifically under the transferring setting, from the perspective of spectral analysis [67] and graphon analysis [68, 69]. Though not directly tackling GFM development, they shed light on developing more effective adaption strategies, which is one potential future direction of our work.

Apart from graph-centric GFM, LLM-induced GFM [70] aims to handle various tasks through the inherent multi-task processing capability of LLMs and leveraging language and next token prediction as a natural medium to unify diverse tasks. Their primary focus is on language-centric tasks with certain graph structures, such as graph-based QA tasks, like the GraphQA dataset [71]. These works, like [72, 32, 71] focus on graph-related QA tasks and adapt existing LLMs to answer a series of questions related to graph structures. Specifically, these models demonstrate task generalization capabilities to unseen tasks. [73] adopts a plugin module to "translate" graphs into text, after which LLMs can answer structure-aware open-ended tasks, including traditional node classification and

GraphQA tasks. [74] goes one step further by unifying all graphs into texts. It then adopts instruction tuning to align LLMs with these graph representations better, enabling them to tackle a wide range of graph-related tasks. Despite the general capability of these models, they still exhibit limited capability on traditional graph machine learning tasks, especially those where structure plays an important role, like link prediction and graph classification.

GLBench [75] is a recently proposed benchmark evaluating both graph language models and large language models on traditional graph machine learning tasks. They mainly focus on node classification task performance on a single dataset, which differs from the cross-graph study adopted in this paper.

## C.2 Learning over Text-attributed Graphs

After unifying diverse features in the text space, the augmented graph naturally becomes a text-attributed graph (TAG), making relevant techniques for TAG applicable to text-space GFMs. The core challenge of learning over TAG lies in integrating node features and graph structures. LLM is adopted from the feature side due to its superior performance in text processing. From the structure side, graph neural networks have become the de facto approach for handling graph-structured data. As a result, the main research objective for learning over TAGs is how to integrate these two models.

One basic approach to address this problem is to cascade the two models, forming a cascading structure [76, 22, 77, 78]. Specifically, embeddings are generated through an LLM, whose parameters are then fixed, followed by training a GNN. To better adapt the LLM to specific data, some works propose using domain adaptive pretraining [76, 77] to generate embeddings more aligned with the downstream task. The drawback of the cascading structure is the tenuous connection between the LLM and GNN, as LLM does not consider the influence of graph structure when generating embeddings. Therefore, structure-aware joint learning has been proposed [79, 80, 81]. [80] introduces a framework for co-training LLM and graph GNN by leveraging each other's generated embeddings. [79] extends this approach by incorporating pseudo-labels generated by both LLMs and GNNs into the optimization process, thus further enhancing the co-training capabilities of the two model types. To better understand the effectiveness of joint learning structure, [33] conducts a benchmark to evaluate different approaches to joint LLM-GNN learning. Despite the claiming superiority of joint learning over cascading structures, experimental results [33, 22, 77] show that with proper LLM selections, cascading structures can achieve better performance with significantly lower computational overhead. This has led to cascading structures becoming the widely adopted design in text-space GFM.

Beyond model-centric research, [21] enhances the effectiveness of learning over TAGs from a data perspective. Specifically, it augments the original node features by generating additional explanation through an LLM. In Section 4.2.1, we observe that co-training has limited improvement on node classification, suggesting that data augmentation may be an effective means further to enhance GFM's performance in node classification tasks. [82] focuses on addressing zero-shot learning on TAGs. Combining the inherent zero-shot capabilities of LLMs with an active learning framework can effectively solve node classification problems on TAGs without manual annotation.

# D Datasets

Details of adopted datasets are presented in Table 11, for the number of edges, we consider all graph as undirected graph and remove all self-loops.

## D.1 Inspection of Node-level and Link-level datasets

In this section, we demonstrate the inspection of text feature similarity of different node-level and link-level datasets.

For the first group of inspection, we compare CS citation graphs, Pubmed, and WikiCS. For the second group of inspection, we compare E-commerce graphs and Amazon ratings.

We then inspect the homophily ratio of each dataset, shown in Table 12.

Finally, we demonstrate the feature space plot in Figure 8.

Table 11: Details of our selected datasets. For Products, we sample a subset of the original dataset since subgraph-based GFM methods are hard to scale to datasets with millions of nodes. Datasets with * are not adopted in co-training and transferring experiments.

| Name | #Graphs | #Nodes | #Edges | Domains | Tasks | #Classes | Metrics |
|------|---------|--------|--------|---------|-------|----------|---------|
| Cora | 1 | 2708 | 10556 | CS Citation | Node, Link | 7 | Accuracy, Hits@100 |
| CiteSeer | 1 | 3186 | 8450 | CS Citation | Node, Link | 6 | Accuracy, Hits@100 |
| Arxiv | 1 | 169343 | 2315598 | CS Citation | Node, Link | 40 | Accuracy, Hits@100 |
| Arxiv23 | 1 | 46198 | 77726 | CS Citation | Node, Link | 40 | Accuracy, Hits@100 |
| History | 1 | 41551 | 503180 | E-commerce | Node, Link | 12 | Accuracy, Hits@100 |
| Child | 1 | 76875 | 2325044 | E-commerce | Node, Link | 24 | Accuracy, Hits@100 |
| Computers | 1 | 87229 | 1256548 | E-commerce | Node, Link | 10 | Accuracy, Hits@100 |
| Photo | 1 | 48362 | 873782 | E-commerce | Node, Link | 12 | Accuracy, Hits@100 |
| Sportsfit | 1 | 173055 | 3020134 | E-commerce | Node, Link | 13 | Accuracy, Hits@100 |
| Products | 1 | 316513 | 19337722 | E-commerce | Node, Link | 39 | Accuracy, Hits@100 |
| Amazon Ratings | 1 | 24492 | 186100 | E-commerce | Node, Link | 5 | Accuracy, Hits@100 |
| Pubmed | 1 | 19717 | 88648 | Bio Citation | Node, Link | 3 | Accuracy, Hits@100 |
| WikiCS | 1 | 11701 | 431726 | Knowledge | Node, Link | 10 | Accuracy, Hits@100 |
| Tolokers(*) | 1 | 11758 | 1038000 | Anomaly | Node, Link | 2 | Accuracy, Hits@100 |
| DBLP(*) | 1 | 14376 | 431326 | CS Citation | Node, Link | 4 | Accuracy, Hits@100 |
| CheMBL | 365065 | 26 | 112 | Biology | Graph | 1048 | Not used for downstream tasks |
| PCBA | 437092 | 26 | 56 | Biology | Graph | 128 | AP |
| HIV | 41127 | 26 | 55 | Biology | Graph | 2 | ROC-AUC |
| Tox21 | 7831 | 19 | 39 | Biology | Graph | 12 | ROC-AUC |
| Bace | 1513 | 34 | 74 | Biology | Graph | 2 | ROC-AUC |
| Bbbp | 2039 | 24 | 52 | Biology | Graph | 2 | ROC-AUC |
| Muv | 93087 | 24 | 53 | Biology | Graph | 17 | ROC-AUC |
| Toxcast | 8575 | 19 | 39 | Biology | Graph | 588 | ROC-AUC |

Table 12: Homophily ratio for node-level datasets

| Dataset | Homophily Ratio |
|---------|-----------------|
| Cora | 0.81 |
| Citeseer | 0.78 |
| Arxiv | 0.66 |
| Arxiv23 | 0.65 |
| History | 0.66 |
| Child | 0.42 |
| Computers | 0.83 |
| Photo | 0.75 |
| Sportsfit | 0.90 |
| Products | 0.81 |
| Pubmed | 0.80 |
| WikiCS | 0.65 |
| Tolokers | 0.59 |
| Amazon ratings | 0.38 |

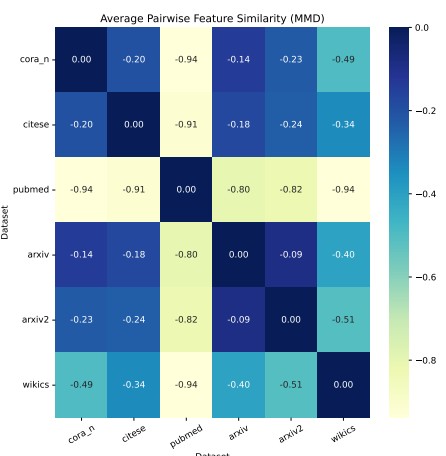

Figure 6: Heatmap of the first set of datasets

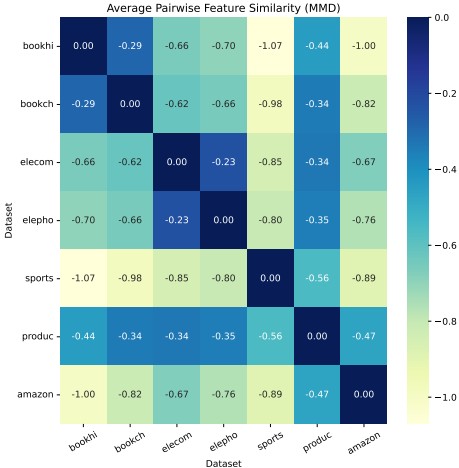

Figure 7: Heatmap of the second set of datasets

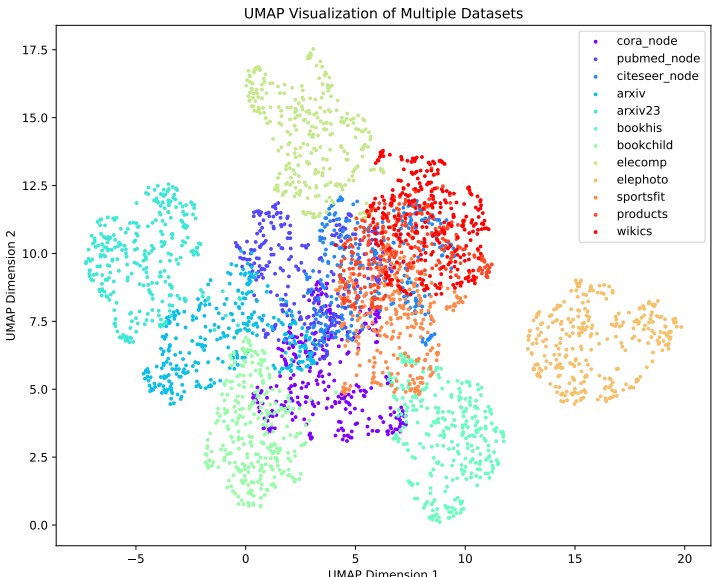

Figure 8: Feature space of the node-level datasets

We don't show the feature space plot for graph-level datasets because their features are all based on prompts for elements that are shared.

## D.2 Dataset Introduction

In this section, we introduce the dataset we use. We need to convert the original node features and labels into natural languages to construct a text-space dataset. We adopt Gemini [36] to generate corresponding descriptions. All datasets mentioned below are under the MIT License unless otherwise specified.

**Cora, CiteSeer, Pubmed.** These datasets are originally adopted in [83]. In the original version, only processed TF-IDF features are provided. So, we follow [21, 22] to extract the original text attributes.

**Arxiv.** This dataset is originally provided in [84]. We adopt the text-space version from [19].

**Arxiv23.** This dataset is originally provided in [21]. The original link is `https://github.com/XiaoxinHe/tape_arxiv_2023`. We then transform it into the text-space dataset.

**History, Child, Computers, Photo, Sportsfit,** These datasets are originally adopted in [33]. They are extracted from [85].

**Products** This dataset is originally provided in [84]. We adopt the original text features provided in the official library. Considering the size of the original dataset, it takes too much time for subgraph-based methods like OneForAll to train on this dataset. As a result, we extract a subgraph with 316513 nodes using torch-geometric's NeighborLoader.

**Amazon Ratings, Tolokers.** These datasets are originally proposed in [35]. Compared to other datasets, these don't follow the commonly adopted homophily assumption for node classification tasks [86]. We crawl the original attributes of these datasets and transform them into texts.

**DBLP.** This dataset is originally proposed in [87]. We consider the paper co-author relationship and turn it into a four-way classification.

**CheMBL, PCBA, HIV, Tox21, Bace, Bbbp, Muv, Toxcast.** These datasets are originally proposed in [34]. Following [19] and [24], we extract the expert prompt to convert the original attributes into texts.

## E   Detailed Experimental Settings

**Computational Environments.**   Our experiments are conducted on a single server with 8 A6000 GPUs.

### E.1   Hyperparameter Settings.

**Co-training.**   The hyper-parameter settings for co-training phase are as follows:

1. For GraphMAE, we use

```
num_heads=4, num_out_heads=1, num_layers=3, num_hidden=1024,
residual=True, in_drop=0.5, attn_drop=0.5, norm='batchnorm',
lr=0.01, weight_decay=1e-05, negative_slope=0.2, activation='prelu',
mask_rate=0.75, drop_edge_rate=0.0, replace_rate=0.2,
scheduler='cosine', warmup=true
```

2. For DGI, we use

```
num_layers=3, num_hidden=512, residual=True, in_drop=0.5, attn_drop=0.5,
    norm='batchnorm', lr=0.001, weight_decay=0.0005, activation='relu',
    scheduler = 'none'
```

3. For OneForAll, we adopt the following set of hyperparameters for node-level and link-level tasks.

```
num_layers=5, num_hidden=384, lr=0.0001, weight_decay=0, JK='none',
    activation='relu'
```

   For graph-level tasks, we set the `num_layers=7`.

4. For LLaGA, we follow the hyper-parameter settings in the original paper [25].

5. For BUDDY and SEAL, we generally follow the hyper-parameter settings in the repo `https://github.com/melifluos/subgraph-sketching`. For BUDDY, the only parameter we tune is the `max_hash_hops`, and we set it to 2 on small-scale graphs Cora, CiteSeer, and Pubmed. We set it to 3 for the rest of the graphs. For SEAL, we set `num_hops` to 2 on small-scale graphs Cora, CiteSeer, and Pubmed. Similarly, we set it to 3 for the rest of the graphs.

For BGRL and GCC, we fail to find a set of hyper-parameters working well for the co-training after searching for a large set of hyperparameters.

**Pre-training.** During the pretraining phase, we adopt hyperparameter settings similar to those in the co-training setup. Therefore, we primarily focus on introducing the relevant settings for pretraining below.

1. For GraphMAE, we pre-train models 10 epochs on the combination of Arxiv, Products, and Sportsfit datasets. Then, we conduct linear probing on downstream tasks.
2. For LLaGA, we pre-train models 1 epoch on the combination of Arxiv, Products, and Sportsfit datasets. Then, we directly output the trained model's prediction for zero-shot inference. For few-shot and fine-tuning cases, we tune the projector with downstream data.
3. For OneForAll, we co-train models on the pre-training datasets for 20 epochs. Then, we directly output the trained model's prediction for zero-shot inference. For few-shot and fine-tuning cases, we tune the models with downstream data.
4. For OneForAll-FS, we pre-train the OneForAll with the few-shot version on the combination of Arxiv, Products, and Sportsfit datasets for 20 epochs. Then, we adopt the trained model for zero-shot and few-shot inference.
5. For Prodigy, we either pre-train it on MAG240M or Arxiv. We construct a 30-way classification problem for both pre-training and generate 20000 randomly selected in-context learning samples.

## F    Implementations

In this paper, we mainly implement the following groups of GFM building blocks. We detail their implementations as follows. All implementations mentioned below are under the MIT License unless otherwise specified. We use a unified data interface for the following methods to pack them for a comprehensive benchmark tool.

**Graph prompts models.** We mainly include two representative text-space models: OneForAll [19] and Prodigy [20]. Their original implementation can be found via `https://github.com/LechengKong/OneForAll` and `https://github.com/snap-stanford/prodigy`. Moreover, we consider two graph prompts designed for pre-training and transferring on the same graph: GPPT [53] and Gprompt [29]. Their implementations are mainly based on `https://github.com/sheldonresearch/ProG`.

**LLM with graph projectors.** We mainly include LLaGA [25] as the baseline for this category for its simplicity and reproducibility. The original implementation can be found via `https://github.com/VITA-Group/LLaGA`. We further include GraphAdapter [54] as a baseline method. Specifically, we adopt GPT2 [88] as the backbone LLMs considering the computation resource restriction. The original implementation can be found from `https://github.com/hxttkl/GraphAdapter`.

**Graph SSL.** We mainly include GraphMAE [42], DGI [39], BGRL [41] as the baseline for this category. For GraphMAE, we follow the implementation from `https://github.com/THUDM/GraphMAE`. For the other baselines, we follow the implementation from PyGCL `https://github.com/PyGCL/PyGCL`.

**Link prediction-specific models.** We mainly include BUDDY [45] and SEAL [46] as two baselines. The original implementation can be found from `https://github.com/melifluos/subgraph-sketching`.

**Pure LLMs.** We also consider two methods purely based on LLMs: GraphLLM [22] and Graph-Text [55]. Their original implementation can be found from `https://github.com/CurryTang/Graph-LLM` and `https://github.com/AndyJZhao/GraphText`.

## G    Extended Experimental Results

### G.1    More results for co-training setting

#### G.1.1    Co-training across node classification and link prediction

**Experiment Settings.** Considering the efficiency of GFMs for link prediction, we adopt three small-scale datasets as in Section 4.2.2. We adopt OneForAll and LLaGA, which can share knowledge between node-level and link-level tasks with a unified model architecture. Since link prediction requires deleting test edges during training, different graph structures are required to evaluate node

Table 13: "Node->Link (Acc)" means removing the test edge and then evaluating link prediction. **-TS** represents "task-specific", which refers to the model trained with a single task. **-CT** represents "cross-task", which refers to the model trained across tasks. Underline represents the case that cross-task co-training benefits compared to task-specific co-training.

| Average performance | OneForAll-TS | LLaGA-TS | OneForAll-CT | LLaGA-CT |
|---|---|---|---|---|
| **Link->Node (Acc)** | 70.57 | 74.65 | 74.03 | 73.45 |
| **Node->Link (Hits@100)** | 74.05 | 85.03 | 79.30 | 84.10 |

classification and link prediction tasks. Therefore, we investigate two cases in which node classification or link prediction is adopted as the downstream task. It should be noted that OneForAll's evaluation in the original paper [19] when co-training node and link-level tasks is potentially problematic since they don't remove the test edges for node-level tasks.

**Results.** As shown in Table 13, we observe that OneForAll achieves positive gain after cross-task co-training compared to task-specific co-training, while LLaGA shows no benefits. For "**node->link**" gain, the possible reason is that link prediction on these datasets requires strong semantic information (as shown in Appendix G.3). In node classification, node features are usually strongly correlated with labels on text-attributed graphs [22]. Therefore, it may help those link prediction tasks requiring strong semantic information. "**Link->Node**" performance gain is probably related to the dataset imbalance issue (a more thorough discussion can be found in Appendix G.5. In node-level datasets, we observe that negative transferring mainly happens on those small-scale datasets (see Table 1). Under co-training, the smaller the amount of data, the more likely it is to be influenced by other datasets. Introducing link prediction is equivalent to adding self-supervision to the same dataset, thereby reducing negative transfer.

**Observation 7.** *Co-training across link prediction and node classification can benefit each other with proper GFM designs.*

### G.1.2 Co-training across node classification, link prediction, and graph classification

**Comprehensive Experiment Settings.** We adopt all datasets from node classification co-training for node-level datasets (Section 4.2.1, three small-scale datasets for link-level datasets (Section 4.2.2), and all datasets from graph classification co-training (Section 4.2.3) for graph-level datasets. We adopt OneForAll, which supports cross-task and cross-graph training. Specifically, we consider the following three cases: (1) Co-training across link-level and graph-level datasets; (2) Co-training across node-level and graph-level datasets; and (3) Co-training across all tasks and datasets. For each dataset, we train the model using the corresponding task. When a dataset contains node and link information, we employ multi-task training, and due to the limited amount of link-level datasets, we focus on node-level performance.

Table 14: Performance of OneForAll on graph classification using MLP and SGC backbone models after conducting node-graph co-training. ST means "single-graph training", CT means "co-training".

| MLP-ST-PCBA | MLP-CT-PCBA | SGC-ST-PCBA | SGC-CT-PCBA |
|---|---|---|---|
| 0.081 | 0.074 | 0.092 | 0.087 |

| MLP-ST-Avg | MLP-CT-Avg | SGC-ST-Avg | SGC-CT-avg |
|---|---|---|---|
| 65.33 | 65.28 | 69.51 | 68.38 |

**Further Probing.** Existing work rarely explores enhancing graph-level task performance through node-level datasets, making our findings somewhat surprising. To better understand this phenomenon, we replace OneForAll's backbone model with MLP and SGC as what we have done for node-level co-training in Section 4.2.1. As shown in Table 14, neither model benefits from co-training this time. This suggests that the positive gain primarily stems from the structural aspect.

When using a GCN backbone and incorporating graph-level co-training, the model tends to learn inductive biases more suitable for graph-level tasks, meaning it makes judgments based on higher-order structures rather than simply relying on augmented features. However, this increased reliance on structure naturally weakens the model's performance in node classification, especially in text-space datasets where features contain strong semantic information.

### G.1.3 Co-training across knowledge graphs and TAGs

In this section, we further study whether adopting knowledge graphs as TAGs can be used to augment the co-training data. We adopt OneForAll as the candidate model and co-train **Cora**, **CiteSeer**, **Pubmed**, with two knowledge graphs, **WN18RR** and **FB15k237**. The experimental results are shown in table 15.

Table 15: Co-training across knowledge graphs and TAGs by viewing knowledge graphs as TAGs.

|  | Cora | CiteSeer | Pubmed | Average |
|---|---|---|---|---|
| **OneForAll(TAG)** | 77.37 | 78.52 | 69.43 | 75.11 |
| **OneForAll(TAG+KG)** | 81.15 | 87.81 | 62.86 | 77.27 |

Preliminary results indicate that co-training with KG can potentially improve model performance on specific datasets (possibly those sharing similarities with KG). At the same time, it might also negatively impact performance on other datasets.

### G.2 Effects of different LLM Encoders

In this section, we further study the influence of different LLM encoders. Due to the size of datasets and computing resource restriction, we limit our scope to medium-scale language model encoder minilm and mpnet [37]. We adopt OFA as the anchor model to compare these two encoders, where the results are shown in Table 16.

The results show that with a more powerful LLM encoder, mpnet, the performance of OneForAll co-trained across node, link, and graph-level tasks clearly improves on node-level tasks.

This result suggests that with the emergence of better LLM encoders, we have reason to believe that stronger LLMs can provide a better feature space, addressing the feature heterogeneity issue across different datasets at the feature level. Using stronger LLM encoders is an effective way to improve node-level performance.

Table 16: Performance comparison of different LLM encoders

| | Cora | CiteSeer | Arxiv | Arxiv23 | History | Child | Photo | Computers | Sports | Products | WikiCS | Pubmed |
|---|---|---|---|---|---|---|---|---|---|---|---|---|
| minilm | 67.55 | 78.37 | 71.79 | 72.96 | 82.96 | 53.39 | 84.5 | 86.32 | 84.5 | 85.58 | 72.16 | 72.59 |
| | pcba | hiv | tox21 | bace | bbbp | muv | toxcast | Node Avg | Graph Avg | | | |
| | 27.9 | 77.69 | 83.25 | 83.23 | 68.14 | 70.78 | 69.79 | 76.06 | 75.48 | | | |
| | Cora | CiteSeer | Arxiv | Arxiv23 | History | Child | Photo | Computers | Sports | Products | WikiCS | Pubmed |
| mpnet | 74.03 | 79.15 | 71.97 | 74.39 | 84.17 | 56.02 | 84.53 | 86.63 | 92.05 | 85.6 | 75.47 | 76.37 |
| | pcba | hiv | tox21 | bace | bbbp | muv | toxcast | Node Avg | Graph Avg | | | |
| | 27.32 | 78.18 | 83.52 | 82.11 | 70.03 | 70.05 | 68.54 | 78.37 | 75.41 | | | |

### G.3 Extended results of co-training over link-prediction tasks

We test more different baseline models in Table 17. Here, each model is trained from scratch on a single graph. The experimental results indicate that a strong correlation exists between node features and ground truth labels for Cora and Citeseer. Even an MLP without structural information can perform well and surpass GCN.

### G.4 Co-training on heterophilous graphs

In Table 2, we observe that OneForAll with an SGC backbone can outperform one with a GCN backbone. However, it should be noted that this only applies to homophilous graphs. Here, we try co-training OneForAll with SGC backbone on graphs from E-commerce and Amazon ratings.

As shown in Table 18, the experimental results demonstrate that while the SGC backbone performs well on datasets conforming to the homophily assumption, its inductive bias does not effectively generalize to heterophilous graphs.

Table 17: Performance of different link prediction backbones trained from scratch on a single graph

|  | Cora | CiteSeer | Pubmed |
|---|---|---|---|
| **MLP** | 83.22 | 91.95 | 49.88 |
| **GCN** | 83.12 | 88.91 | 66.14 |
| **SIGN** | 87.77 | 84.73 | 43.12 |
| **BUDDY** | 91.37 | 96.57 | 83.29 |

Table 18: Performance of OneForAll after co-training on graphs from E-commerce and Amazon Ratings.

|  | History | Child | Photo | Computers | Sports | Products | Ratings |
|---|---|---|---|---|---|---|---|
| GCN-backbone | 83.3 | 56.22 | 85.05 | 87.83 | 92.29 | 86.91 | 48.21 |
| SGC-backbone | 84.2 | 56.6 | 85.8 | 87.7 | 91.8 | 87.1 | 43.88 |

### G.5 Effects of dataset scales on co-training

In OneForAll, the authors address the issue of negative transfer by adding weights to different datasets. This raises the question: is the negative transfer between datasets caused by dataset imbalance? To answer this, we first focus on the E-commerce dataset, which doesn't exhibit significant negative transfer. Unlike the original split, we adopt the same low labeling rate as in Cora and Citeseer for co-training, with 20 samples per class in the training set.

As shown in Table 19, we observe that the occurrence of negative transfer appears to be unrelated to dataset ratio but rather depends on the inherent characteristics of the datasets. Interestingly, datasets within the E-commerce domain are closer in feature space compared to those in the CS citation domain, yet we observe better transferability between the former.

Table 19: In this table, we change every dataset's training ratio to 20 samples per class. Unless Table , we use fixed hyper-parameter setting for single-graph training.

|  | History | Child | Photo | Computers | Sports | Products | Avg |
|---|---|---|---|---|---|---|---|
| **Single** | 69.10 | 22.70 | 61.10 | 63.30 | 57.50 | 61.30 | 55.83 |
| **Co-train** | 61.60 | 25.90 | 62.10 | 58.30 | 67.20 | 63.50 | 56.43 |
|  | **Cora** | **Citeseer** | **Arxiv** | **Arxiv23** | **Avg** |  |  |
| **Single** | 80.13 | 81.35 | 59.40 | 58.30 | 69.80 |  |  |
| **Co-train** | 60.10 | 82.10 | 55.30 | 47.90 | 61.35 |  |  |

It's worth noting that we also notice that when each individual dataset has a sufficient number of data points, negative transfer rarely occurs. For instance, we try removing all small-scale datasets from the node-level experiments, and the co-training results are as Table 20.

From the table, We observe that even though these datasets come from diverse domains, negative transfer does not occur.

## H  Limitations and future works

**Limitations of experiments.**    Due to computational constraints, we do not utilize multiple seeds to reduce experimental variance in most experiments, as a single full training run of models like OneForAll and LLaGA takes several days. Conducting multiple trials is a potential avenue for future work.

**Limitations of scopes.**    In this paper, we primarily focus on the issue of feature heterogeneity. To address existing structural heterogeneity, we mainly adopt current solutions within GFM, such

Table 20

| | Arxiv | Arxiv23 | History | Child | Photo | Computers | Sports | Products | Avg |
|---|---|---|---|---|---|---|---|---|---|
| **Single** | 73.85 | 73.75 | 83.33 | 53.77 | 84.46 | 86.48 | 92.50 | 87.35 | 79.44 |
| **Co-train** | 72.50 | 73.40 | 83.42 | 55.99 | 84.67 | 87.39 | 92.23 | 86.84 | 79.56 |

as directly employing the classic MPNN. The effectiveness of newer architectures, like graph transformers [60, 89], and their ability to generalize across diverse tasks remain open questions for further investigation. In this paper, we primarily focus on the co-training setting. To transfer a co-trained model to new data, there are two possible approaches: 1. Incorporating the new dataset in a co-training manner requires retraining the model, making it not feasible. 2. Directly applying or fine-tuning the co-trained model on the new data. Our experimental results show that GFMs haven't demonstrated promising results in this setting, making it a crucial challenge for the next phase. As future work, we may explore the transferring methods mentioned in [68] and compare them with approaches listed in our papers.

**Potential future directions.** We showcase some potential applications based on our proposed benchmarks:

1. **A comprehensive analysis of the cross-dataset neural scaling capabilities of GNN models**: Our benchmark, through a unified feature space, can be used to study the neural scaling properties of GNNs and graph self-supervised learning models.
2. **Developing new methods for cross-dataset alignment**: Research how to co-train models on multiple datasets to achieve better performance; for example, we can test the effectiveness of [90] on our proposed datasets.
3. **Foundation model for link prediction**: The benchmark's empirical observations show that co-training significantly improves link prediction performance when models present proper inductive biases. Thus, a foundation model for link prediction is promising and can potentially lead to large improvements in OGB datasets [84].
4. **Developing models for text-attributed graphs**: Since our datasets, all have text attributes and text descriptions of labels, they can be used to study related techniques. We will add a section to discuss the usage and design of this benchmark.

# I   Broader Impacts

In this paper, we provide empirical investigation for the development of graph foundation models, which may empower diverse applications including E-commerce, social network, and natural science. GFM has the potential to significantly reduce the resource consumption associated with training numerous task-specific models. Additionally, it can drastically minimize the need for manual annotation, especially in domains like molecular property prediction. We believe our contributions will accelerate ongoing efforts to develop the next generation of versatile and equitable graph foundation models.

A potential negative impact of GFM is that due to its unified backbone pre-trained on massive data, some popular biases reflected in the datasets may be present in GFM's predictions, which requires user attention.

