# Text-space Graph Foundation Models: Comprehensive Benchmarks and New Insights: Supplementary Information

**Zhikai Chen[1], Haitao Mao[1], Jingzhe Liu[1], Yu Song[1], Bingheng Li[1],**
**Wei Jin[2], Bahare Fatemi[3], Anton Tsitsulin[3], Bryan Perozzi[3],**
**Hui Liu[1], Jiliang Tang[1]**
[1]Michigan State University, [2]Emory University, [3]Google Research

## 1   Overview of Supplementary Information

In this section, we will first briefly summarize the content included in the supplementary information. The technical appendices in the main text have already covered most of the experimental details, such as the computation environment, hyperparameter settings, and the introduction of the datasets. Therefore, we will mainly provide a more comprehensive datasheet and instructions for code reproduction based on the original appendices.

## 2   Datasheet

### 2.1   Motivation

1. **For what purpose was the dataset created?** *Our benchmark dataset was created to serve as a foundation for evaluating the effectiveness of text-space graph foundation models (GFMs) across diverse domains of data and tasks.*

2. **Who created the dataset and on behalf of which entity?** *The dataset was developed by ML researchers listed in the author list.*

3. **Who funded the creation of the dataset?** *Funding sources of authors will be listed in the acknowledgment section of the main text.*

### 2.2   Distribution

1. **Will the dataset be distributed to third parties outside of the entity (e.g., company, institution, organization) on behalf of which the dataset was created?** *Yes, the dataset is open to the public.*

2. **How will the dataset will be distributed (e.g., tarball on website, API, GitHub)?** *The dataset will be distributed through Huggingface* `https://huggingface.co/datasets/zkchen/tsgfm`*, and the code will be used for developing baseline models through GitHub.*

3. **Have any third parties imposed IP-based or other restrictions on the data associated with the instances?** *No.*

4. **Do any export controls or other regulatory restrictions apply to the dataset or to individual instances?** *No.*

### 2.3 Maintenance

1. **Who will be supporting/hosting/maintaining the dataset?** *DSE Lab from Michigan State University will support, host, and maintain the dataset.*

2. **How can the owner/curator/manager of the dataset be contacted (e.g., email address)?** *The owner/curator/manager(s) of the dataset can be contacted through the following emails: Zhikai Chen(chenzh85@msu.edu)*

3. **Is there an erratum?** *No. If errors are found in the future, we will release errata on the project's GitHub page `https: // github. com/ CurryTang/ TSGFM`.*

4. **Will the dataset be updated (e.g., to correct labeling errors, add new instances, delete instances)?** *Yes, the datasets will be updated as needed to ensure accuracy. Announcements regarding updates will be posted on the project's main webpage `https: // github. com/ CurryTang/ TSGFM`.*

5. **If the dataset relates to people, are there applicable limits on the retention of the data associated with the instances (e.g., were the individuals in question told that their data would be retained for a fixed period of time and then deleted?)** *N/A*

6. **Will older version of the dataset continue to be supported/hosted/maintained?** *Yes, older versions of the dataset will continue to be maintained and hosted.*

7. **If others want to extend/augment/build on/contribute to the dataset, is there a mechanisms for them to do so?** *Yes. They can submit a pull request on the Github page.*

### 2.4 Composition

1. **What do the instance that comprise the dataset represent (e.g., documents, photos, people, countries?)** *Each instance includes a Pytorch-geometric [1] like data object. It contains vector objects to store the structural and feature information of the graph dataset.*

2. **How many instances are there in total (of each type, if appropriate)?** *We include 23 datasets in this benchmark. The detailed statistics of each one can be found in the main text or the Github page.*

3. **Does the dataset contain all possible instances or is it a sample of instances from a larger set?** *Yes.*

4. **Is there a label or target associated with each instance?** *Yes.*

5. **Is any information missing from individual instances?** *No.*

6. **Are there recommended data splits (e.g., training, development/validation, testing)?** *We write a split function in the code of the project. The split is deterministic if the random seed is fixed.*

7. **Are there any errors, sources of noise, or redundancies in the dataset?** *There may exist potential errors due to the annotation bias in the labeling process of the original dataset.*

8. **Is the dataset self-contained, or does it link to or otherwise rely on external resources (e.g., websites, tweets, other datasets)?** *The dataset is self-contained.*

9. **Does the dataset contain data that might be considered confidential?** *No.*

10. **Does the dataset contain data that, if viewed directly, might be offensive, insulting, threatening, or might otherwise cause anxiety?** *No.*

### 2.5 Collection Process

1. **How was the data associated with each instance acquired?** *The data is collected from the raw version including [2, 3, 4, 5, 6, 7] by either crawling the raw texts or generating the corresponding text features. Detailed data processing functions can be seen on the GitHub page.*

2. **What mechanisms or procedures were used to collect the data (e.g., hardware apparatus or sensor, manual human curation, software program, software API)?** *Python code.*

3. **Who was involved in the data collection process (e.g., students, crowdworkers, contractors) and how were they compensated (e.g., how much were crowdworkers paid)?** *Listed authors.*

78  4. **Does the dataset relate to people?** *No.*

79  5. **Did you collect the data from the individuals in questions directly, or obtain it via third**
80  **parties or other sources (e.g., websites)?** *Third parties.*

81 ### 2.6 Uses

82  1. **Has the dataset been used for any tasks already?** *No, this dataset has not been used for*
83  *any tasks yet.*

84  2. **What (other) tasks could be the dataset be used for?** *These datasets can be used for any*
85  *tasks related to graph machine learning.*

86  3. **Is there anything about the composition of the dataset or the way it was collected and**
87  **preprocessed/cleaned/labeled that might impact future uses?** *No.*

88  4. **Are there tasks for which the dataset should not be used?** *No.*

89 ### 2.7 Accessibility

90 The datasets can be downloaded from `https://huggingface.co/datasets/zkchen/tsgfm` and
91 the code can be accessed from `https://github.com/CurryTang/TSGFM`. The DOI of our dataset
92 is `10.57967/hf/2455`.

93 ## 3 Instructions for reproducibility

94 Instructions for reproducing the benchmark results can be found in the readme of the Github Reposi-
95 tory `https://github.com/CurryTang/TSGFM`.

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

116    Benchmarking and rethinking. Advances in Neural Information Processing Systems, 36:17238–
117    17264, 2023. Cited on page 2.