# OpenReview forum: "Text-space Graph Foundation Models: Comprehensive Benchmarks and New Insights"
_NeurIPS.cc/2024/Datasets_and_Benchmarks_Track — NeurIPS 2024 Track Datasets and Benchmarks Poster_

### Official Review · Reviewer_SB91 · 2024-06-26

**Rating:** 5
**Confidence:** 3
**Clarity:** Good.

**Review:**

**Pros**

1. **Diverse Text-Space Datasets:** The paper compiles a wide range of diverse text-space datasets from various domains such as academic, e-commerce, and biology. This comprehensive collection is crucial for thorough evaluation and benchmarking of Graph Foundation Models (GFMs), ensuring that the models are tested against varied and representative data.
2. **Experimental Results Across Different Scenarios:** The paper provides experimental results across various scenarios using these diverse text-space datasets. This evaluation offers certain valuable insights into the performance and effectiveness of GFMs in various settings, showing their practical applicability and limitations.
3. **Complete Pipeline Implementation:** The paper implements a complete pipeline that includes GraphSSL, foundational graph prompt models, and LLM with graph projectors. This robust framework demonstrates a practical approach to evaluating and improving GFMs, showcasing the integration of multiple advanced techniques into a cohesive system.

**Cons**

1. **Lack of Evaluation of Real LLMs on Graph Foundation Models:** The paper does not assess the performance of real large language models (LLMs) like LLaMA on graph foundation models, which is an important aspect that needs exploration.
2. **Emphasis on Co-Training Over Pre-Training:** While addressing cross-tasks and cross-datasets problems in GFMs is closely related to the pre-training process, the paper devotes a significant portion to discussing co-training results. *A stronger focus on pre-training would have been more relevant and impactful.*
3. **Missing Baselines:** The paper lacks certain baseline models, such as GraphGPT [1] or GraphText [2], which are important for a comprehensive evaluation.
4. **Incomplete Baselines for Graph Classification and Link Prediction:** The baselines are not comprehensive enough, as the paper does not include comparisons with relevant graph SSL methods for graph classification and link prediction tasks, such as GraphCL [3], EdgePredGPPT [4], and EdgePreGPrompt [5]. This makes it difficult to judge the upper limits and advantages of the proposed models.

[1] Tang, Jiabin, et al. "Graphgpt: Graph instruction tuning for large language models." arXiv preprint arXiv:2310.13023 (2023).

[2] Zhao, Jianan, et al. "Graphtext: Graph reasoning in text space." arXiv preprint arXiv:2310.01089 (2023).

[3] You, Yuning, et al. "Graph contrastive learning with augmentations." Advances in neural information processing systems 33 (2020): 5812-5823.

[4] Sun, Mingchen, et al. "Gppt: Graph pre-training and prompt tuning to generalize graph neural networks." Proceedings of the 28th ACM SIGKDD Conference on Knowledge Discovery and Data Mining. 2022.

[5] Liu, Zemin, et al. "Graphprompt: Unifying pre-training and downstream tasks for graph neural networks." Proceedings of the ACM Web Conference 2023. 2023.

**Strengths:**

See the previous content.

**Additional Feedback:**

Not applicable.

**Correctness:**

The shown experimental results look sound and the implementation details can be referred to the given link: https://github.com/CurryTang/TSGFM

**Documentation:**

Not applicable.

**Ethics:**

Not applicable.

**Limitations:**

Not applicable.

**Opportunities For Improvement:**

Referring to the previous cons.
Concretely,
1. Reorganize the paper structure, focusing on the graph pre-training phase.
2. Supplement more graph LLM models as much as possible, such as GraphGPT, GraphText, and GraphAdapter [1].
3. Include more graph SSL methods, such as graph-level GraphCL and edge-level EdgePredGPPT and EdgePredGprompt.

**Relation To Prior Work:**

Yes.

**Summary And Contributions:**

The paper "Text-space Graph Foundation Models: Comprehensive Benchmarks and New Insights" introduces a novel benchmark to evaluate and enhance text-space Graph Foundation Models (GFMs).

Here are the Contributions:

1. **Novel Text-Space Datasets:** The authors curated and preprocessed over 20 datasets spanning various domains including academic, e-commerce, and biology to address the scarcity of existing text-space datasets and evaluation based on text space.
2. **Comprehensive Evaluation on Diverse Use Cases:** The authors defined four applicable GFM paradigms and evaluated different GFM building blocks under each setting, providing a more comprehensive GFM setting than existing works.
3. **Novel Insights:** Empirical results provided new insights, revealing that although LLMs offer a promising initial performance, gaps still exist across different datasets. Positive transfer in text-space GFMs relies on transferable structural patterns combined with appropriate inductive biases for downstream tasks.

---

> ### Author Rebuttal · Authors · 2024-08-17
>
> Thanks for your valuable feedback.
> > Q1. Lack of Evaluation of ...
>
> A1. First, we evaluate an LLM-based GFM, LLaGA, using Mistral-7B as the backbone and transforming the embedding output by the linear GNN into the text space. We acknowledge that other LLM-based baselines should also be compared in the transferring experiments. Therefore, we have added new baselines: GraphAdapter[1], Graphtext[2], GraphLLM[3], GraphGPT[4], GPPT[5], GraphPrompt[9] with results shown below.
>
> It should be noted that we considered GraphGPT and GraphAdapter when preparing this paper. However, we fail to train them from scratch because of efficiency issues. To conduct the experiment here, similar to [6], we adopt the official checkpoint of GraphGPT for the zero-shot setting. For GraphAdapter, adopting a 7B model will generate TB-scale intermediate data beyond our server's capability. So, we adopt GPT2-large as the backbone model.
>
>
> |  | Cora |  |  | History |  |  | Ratings |  |  |
> |:---:|:---:|:---:|:---:|:---:|:---:|:---:|:---:|:---:|:---:|
> |  | 0 shot | 3 shot | FT | 0 shot | 3 shot | FT | 0 shot | 3 shot | FT |
> | LLaGA | 18.25 | 60.7 | 80.45 | 22.05 | 36.45 | 82.55 | 23.15 | 23.45 | 28.2 |
> | Prodigy | N/A | 71.60 | N/A | N/A | 52.66 | N/A | N/A | 67.41 | N/A |
> | Simple SBERT | 67.41 | 68.42 | 82.2 | 59.25 | 51.25 | 85.3 | 27.39 | 20.95 | 48.46 |
> | GraphText | N/A | 50.33 | N/A | N/A | 48.00 | N/A | N/A | 37.67 | N/A |
> | GraphAdapter | 21.09 | 33.74 | 62.56 | 17.43 | 36.60 | 82.94 | 27.64  | 29.41 | 38.69 |
> | GraphLLM | 67.33 | 68.00 | N/A | 34.67 | 56.67 | N/A | 24.33 | 34.33 | N/A |
> | GraphGPT | 22.50 | N/A | N/A | 15.50 | N/A | N/A | 24.00 | N/A | N/A |
> | GPPT | N/A | 44.14 |	65.84 | N/A | 27.54 | 35.94 | N/A | 14.24 | 20.22 |
> | GPrompt | N/A | 55.38 | 70.82 | N/A |17.36 |21.33 | N/A |15.38 |17.23 |
>
> The extra experimental results demonstrate the following conclusions:
> * For few-shot inference, two strategies are shown to be effective: 1. LLM embedding with proper graph inductive bias, and then the task should be reformulated to find the closet class labels based on similarity; 2. Pure LLM with text inputs (projector doesn't work), and the downstream task should be correlated to the node feature semantics (for ratings, it's a rating prediction task).
> * Semantic label embeddings are critical to the transferability of graph prompts. Otherwise, they fail like GPPT and GPrompt.
>
> > Q2. Emphasis on Co-Training Over Pretraining: ...
>
> A2. We acknowledge that the original manuscript's background introduction and emphasis on the transferring setting are insufficient. In the revision, we will address this by 1. Adding a more comprehensive introduction to the transferring problem setting and 2. Incorporating additional baselines like GraphLLM, Graphtext, GraphAdapter, and GPPT. The results are shown in our answer to question 1.
>
> We focus on co-training settings for the following reasons: 1. GFM development is still in its early stages, and our main goal is to understand the potential unifying features in the text space to develop GFMs. We aim to derive principles from experimental results to guide future model design. In this sense, co-training better suits our needs since they can directly reflect the model's performance while transferring also involves adapting pre-trained models to downstream tasks, which can be considered a further step based on effective co-training. 2. Co-training is an important ingredient for efficient transferring [7]. Considering the limited size of a single graph, it's necessary to co-train them together to scale models.
>
> > Q3. Missing Baselines: The paper lacks certain baseline models, such as ...
>
> A3. We present the experimental results for GraphGPT, GraphText, GraphLLM, and GraphAdapter in our answer to question 1.
>
> We also want to clarify our criteria for selecting models. We aim to study the properties and underlying principles exhibited by graph models as they evolve from single-dataset single-task settings to multi-dataset multi-task settings, inspiring the further design of GFMs. To achieve this, we choose representative models that do not require particular training strategies, such as multi-stage training, and have simple designs to facilitate comparative experiments. Our benchmark differs in design philosophy from [6], which focuses on testing the performance of different models on individual datasets. We will add discussion to those parallel works in the revision.

---

> > ### Author Rebuttal · Authors · 2024-08-17
> >
> > > Q4. Incomplete Baselines for Graph Classification and...
> >
> > A4. We present the experiments for GraphCL [8], GPPT [5], GPrompt [9], and GPF [10] in the following table. GPPT and GPrompt are usually not evaluated on link prediction tasks since there's no need to generate class embeddings. We check them on node and graph classification tasks.
> >
> > We first apply them to the graph classification task. We compare the following two ways to use GraphCL: 1. Pre-training it on ZINC with original chemical features and then fine-tuning; 2. Training and fine-tuning it on downstream task data. For GPPT and Gprompt, we also consider pre-training on ZINC. For methods pre-trained on ZINC, the performance on PCBA is not available considering the duplication between PCBA and ZINC.
> >
> > |  | pcba | hiv | tox21 | bace | bbbp | muv | toxcast |
> > |---|---|---|---|---|---|---|---|
> > | OneForAll | 23.61 | 75.24 | 82.5 | 77.32 | 69.97 | 70.39 | 68.39 |
> > | GraphCL (co-train) | 20.26 | 70.33 | 73.74 | 64.42 | 63.09 | 73.62 | 60.77 |
> > | GraphCL (Zinc, transferring) | N/A | 78.47 | 73.87 | 75.38 | 69.68 | 69.8 | 62.4 |
> > | EdgePred-GPPT | N/A | 73.19 | 78.91 | 70.31 | 69.43 | 82.06 | 64.86 |
> > | EdgePred-GPrompt | N/A | 59.31 | 68.09 | 67.70 | 69.29 | 62.35 | 60.54 |
> > | GPF | N/A | 77.60 | 79.74 | 81.57 | 69.57 | 82.86 | 65.65 |
> >
> > From the experimental results, we have the following observations:
> > 1. If we directly compare the co-training setting (pre-train and finetune on the same set of data), supervised methods like OneForAll present clear benefits, which can be attributed to unifying the label space.
> > 2. GraphSSL and graph prompt methods are effective when pre-training on large-scale external data (like ZINC), and methods like GPF with learnable token embedding show the best performance.
> > 3. The trade-off between few-shot and fine-tuning performance is a challenge here. GPrompt reformulates the task formulation as finding closet labels in the embedding space and thus can not utilize labels as well as traditional methods.
> >
> > We further apply GPPT and GPrompt to the transferring setting and compare them with other SSL methods like GraphMAE and OneForAll.
> >
> > |  | Cora | CiteSeer |
> > |---|---|---|
> > | OneForAll | 70.74 | 81.66 |
> > | GraphMAE | 78.09 | 68.8 |
> > | GPPT | 70.92 | 63.38 |
> > | Gprompt | 68.13 | 54.95 |
> >
> > We find that prompt-based models GPPT and GPrompt show limited effectiveness. The main reason is that they are designed based on numeric labels instead of semantic label embeddings. The former doesn't present good transferability across datasets.
> >
> >
> > [1] Huang, Xuanwen, et al. "Can GNN be Good Adapter for LLMs?." Proceedings of the ACM on Web Conference 2024. 2024.
> >
> >
> > [2] Zhao, Jianan, et al. "Graphtext: Graph reasoning in text space." arXiv preprint arXiv:2310.01089 (2023).
> >
> > [3] Chen, Zhikai, et al. "Exploring the potential of large language models (llms) in learning on graphs." ACM SIGKDD Explorations Newsletter 25.2 (2024): 42-61.
> >
> > [4] Tang, Jiabin, et al. "Graphgpt: Graph instruction tuning for large language models." arXiv preprint arXiv:2310.13023 (2023).
> >
> > [5] Sun, Mingchen, et al. "Gppt: Graph pretraining and prompt tuning to generalize graph neural networks." Proceedings of the 28th ACM SIGKDD Conference on Knowledge Discovery and Data Mining. 2022.
> >
> > [6] Li, Yuhan, et al. "GLBench: A Comprehensive Benchmark for Graph with Large Language Models." arXiv preprint arXiv:2407.07457 (2024).
> >
> > [7] Li, Yuhan, et al. "ZeroG: Investigating Cross-dataset Zero-shot Transferability in Graphs." arXiv preprint arXiv:2402.11235 (2024).
> >
> > [8] You, Yuning, et al. "Graph contrastive learning with augmentations." Advances in neural information processing systems 33 (2020): 5812-5823.
> >
> > [9] Liu, Zemin, et al. "Graphprompt: Unifying pretraining and downstream tasks for graph neural networks." Proceedings of the ACM Web Conference 2023. 2023.
> >
> > [10] Fang, Taoran, et al. "Universal prompt tuning for graph neural networks." Advances in Neural Information Processing Systems 36 (2024).

---

> ### Author Response · Authors · 2024-08-27
>
> Dear reviewers,
>
> We sincerely appreciate your efforts in reviewing this paper and your valuable feedback. We hope our responses have addressed your concerns. If you have any additional questions, please let us know. We look forward to hearing from you.

---

> ### Author Response · Authors · 2024-08-28
> **A friendly reminder**
>
> Dear reviewer SB91,
>
> Thanks for your constructive reviews and efforts. As we approach the end of the discussion period, we would greatly appreciate your feedback. We hope that the responses have addressed your concerns. If there are any further questions, please let us know. We look forward to hearing from you.

---

> > ### Comment · Reviewer_SB91 · 2024-09-04
> >
> > After the rebuttal, the authors have addressed most of my concerns, including the introduction of baselines and the clarification of the co-training setting. However, my main remaining concern is whether the authors can effectively revise the introduction in the final version to emphasize the importance of the transferring problem discussed throughout the paper. Although the authors have mentioned some plans for revision, I find these a bit too broad, and I am not confident that the revised introduction will be clear. Specifically, for readers who are not deeply familiar with GFM, their first impression of the paper might be that it should provide a comprehensive table evaluating existing GFM models on different datasets. However, the main focus of the paper is actually on co-training settings, which is not thoroughly mentioned or discussed in the title or the introduction.
> >
> > From my perspective, it is indeed a good paper but needs to be carefully revised for more readable. Overall, I decide to maintain my score.

---

> > > ### Author Response · Authors · 2024-09-04
> > >
> > > First, we want to thank you for your acknowledgment of our paper and our rebuttal, which addresses most of your concerns.
> > > > My main remaining concern is whether the authors can effectively revise the introduction in the final version to emphasize the importance of the transferring problem discussed throughout the paper.
> > >
> > > We admit that we don't comprehensively discuss the transferring setting in the original submission. We believe this issue can be addressed through the specific revision plan we propose below and the supplementary experiments provided in our previous rebuttal.
> > >
> > > **Revision 1.** Clarifying and pointing out the limitation of our scope. This part will be put in the introduction.
> > >
> > > **Scope of the paper.** Our paper investigates two usage scenarios for GFMs: co-training and transferring, and focuses on the co-training phase due to: (1) the efficiency issue of most current GFMs limits large-scale pre-training/transferring research, hindering rigorous conclusions. Co-training on downstream data and comparing it with model training from scratch can help us better compare the potential of text space models. (2) Achieving improved performance via co-training is a more realizable goal in the current stage. Transferring to new data involves problems like catastrophic forgetting[1] and distribution shift, which is more challenging.
> > >
> > > **Revision 2.** Introducing existing strategies for transferring and why transferring is more challenging than the co-training setting. This will be appended to Section 2.
> > >
> > > Compared to co-training models and then applying them to graphs (sampled) from the same datasets, transferring presents a more challenging scenario due to the distribution gap between pre-training and downstream data.
> > > Currently, transferring mainly involves the following techniques:
> > > 1. Self-supervised or supervised pre-training with task-specific head: This is the traditional method for transferring, which requires a head due to feature and task misalignment. It assumes knowledge stored in the backbone is useful for downstream tasks.
> > > 2. Foundational graph prompts: Similar to co-training, graph prompts unify pre-training and downstream tasks, allowing direct application without tuning heads, enabling in-context learning.
> > > Besides these, actively selecting pre-training data is a common strategy[2], but it requires augmenting pre-training data for specific downstream tasks, conflicting with the GFM setting, so it's not considered within the transferring setting. Transferring and co-training use similar techniques but require generalization to new data and tasks, making it more challenging.
> > >
> > > **Revision 3.** Limitation of our work. This will be appended to Section H.
> > >
> > > In this paper, we primarily focus on the co-training setting. To transfer a co-trained model to new data, there are two possible approaches:
> > > 1. Incorporating the new dataset in a co-training manner, which requires retraining the model, making it not feasible.
> > > 2. Directly applying or fine-tuning the co-trained model on the new data. Our experimental results show that GFMs haven't demonstrated promising results in this setting, making it a crucial challenge for the next phase.
> > >
> > > As future work, we may explore the transferring methods mentioned in [3] and compare them with approaches listed in our papers.
> > >
> > > > Their first impression of the paper might be that it should provide a comprehensive table evaluating existing GFM models on different datasets.
> > >
> > > We want to emphasize that the Text space GFM proposed here is a general paradigm, with implementation methods like graph SSL, graph prompting, and graph LLMs. Our benchmark tests these paradigms within the GFM setting and leads to valuable observations, which differs from applying foundation models to the graph domain and testing them in traditional settings.
> > >
> > > Finally, we want to point out that our study reveals valuable observations for the transferring setting of the GFM:
> > > 1. The superior performance of current model's zero-shot capability mainly comes from the LLM embeddings
> > > 2. On most homophilous graphs, the benefit brought by GFM's pre-training and transferring is limited since SGC-like feature propagation is already a well-transferrable pattern. However, for heterophilous graphs, the benefit of pre-training is clear.
> > >
> > > [1] Liu, Huihui, Yiding Yang, and Xinchao Wang. "Overcoming catastrophic forgetting in graph neural networks." Proceedings of the AAAI conference on artificial intelligence. Vol. 35. No. 10. 2021.
> > >
> > > [2] Xu, Jiarong, et al. "Better with less: A data-active perspective on pretraining graph neural networks." Advances in Neural Information Processing Systems 36 (2023): 56946-56978.
> > >
> > > [3] Sun, Yifei, et al. "Fine-Tuning Graph Neural Networks by Preserving Graph Generative Patterns." Proceedings of the AAAI Conference on Artificial Intelligence. Vol. 38. No. 8. 2024.

---

### Official Review · Reviewer_9PU3 · 2024-07-24
**A good benchmark with interesting observations; how it could benefit the community could be further clarified**

**Rating:** 6
**Confidence:** 4
**Correctness:** Yes
**Clarity:** Yes

**Review:**

Strength:
(1)    The graph foundation model is a trending and important direction for both academia and industry. Text-space models are one potentially promising path.
(2)    The authors have conducted several experiments with some interesting observations.


Weaknesses and Suggestions:
(1)    Though I appreciate the authors have drawn some empirical observations, the current paper is more like an experimental report. The authors could further revise the design and usage side, e.g., why the proposed setting/protocols are most appropriate for text-space GFM, how users can utilize the dataset or customize it for their own usage, etc., so that the benchmark can truly advance the field and benefit the community, like the well-known Open Graph Benchmark.
(2)    Though the current datasets span several domains, their sizes are mostly small or medium-scale. More large-scale datasets, e.g., graphs with billions of nodes and edges, which are available such as larger citation graphs, web page graphs, social networks, etc., can further strengthen the benchmark.
(3)    Considering that this research direction is fast developing, the authors may provide more discussions on whether their observations may hold in the future or be helpful for the community, e.g., whether it would be possible that some sort of text-space graph foundation model may arise that violates all these observations?

**Strengths:**

See above

**Additional Feedback:**

NA

**Documentation:**

Could be improved

**Limitations:**

Yes

**Opportunities For Improvement:**

See above

**Relation To Prior Work:**

Yes

**Summary And Contributions:**

In this paper, the authors propose a benchmark for text-space graph foundation models, i.e., for text-attributed graphs or converting non-text-attributed graphs into textual descriptions, together with empirical analyses. Specifically, the authors adopt 20 datasets spanning from academic citation, E-commerce, biology, etc. Then, the authors categorize the learning paradigms into 4 groups, i.e., co-training and pre-training, task-specific or cross-task. Several empirical observations are also drawn from experiments, e.g., gaps exist across datasets, positive transfer only holds with appropriate inductive biases, etc.

---

> ### Author Rebuttal · Authors · 2024-08-17
>
> Thanks for your valuable feedback.
> > Q1. Though I appreciate ...
>
> A1. We are willing to clarify our benchmark's design philosophy and potential applications.
>
> Regarding the design philosophy, our primary goal is to understand how GFMs (or, more generally, GNNs) behave when scaling with more datasets on a unified feature space. Toward this goal, we study their behavior in two important directions: 1. improving downstream task performance with more data (co-training) and 2. adapting a pre-trained model to a new downstream task (transferring). These two settings are critical to the development of GFM in that:
> * Co-training is a natural extension of the traditional graph machine learning on a single dataset; it's critical to understand the neural scaling properties across datasets and also the generalizable principles [1] (for example, the empirical observations in this paper)
> * Transferring (including fine-tuning, zero-shot, and few-shot learning) is a common paradigm for applying foundation models for other domains, such as LLMs. It can effectively address challenges like cold start in industrial applications, offering high practical value.
>
> These two settings encompass most application scenarios for GFMs, and we also consider the three most common task forms on graphs, making them highly representative.
>
> In terms of the potential applications, we showcase some promising future directions based on our benchmarks:
> 1. A comprehensive analysis of the cross-dataset neural scaling capabilities of GNN models: Our benchmark, through a unified feature space, can be used to study the neural scaling properties of GNNs and graph self-supervised learning models.
> 2. Developing new methods for cross-dataset alignment: Research how to co-train models on multiple datasets to achieve better performance; for example, we can test the effectiveness of [2] on our proposed datasets.
> 3. Foundation model for link prediction: The benchmark's empirical observations show that co-training significantly improves link prediction performance when models present proper inductive biases. Thus, a foundation model for link prediction is promising and can potentially lead to large improvements in datasets like OGBL-Citation2.
> 4. Developing models for text-attributed graphs: Since our datasets all have text attributes and text descriptions of labels, they can be used to study related techniques.
> We will add a section to discuss the usage and design of this benchmark.
>
>
>
>
> > Q2. Though the current datasets span several domains...
>
> A2. We choose small and medium-sized datasets because most current GFMs have specific efficiency issues. We can only evaluate them on medium-scale datasets. For example, training OneForAll once on the current scale of data already takes more than 20 hours. Although Prodigy is trained on the MAG240M dataset, it only samples tens of thousands of subgraphs, so it cannot truly utilize large-scale data. While the versions we provide are mainly medium-sized, we have also collected the original million-scale datasets like Products and Citation2, which can be used to evaluate more efficient methods. Another issue is that we are studying text-space datasets with a unified space. Although they don't necessarily have to be text attributes, they need features that can be converted to text. Therefore, many datasets on NetworkRepository[3] without features are also not applicable. Finally, our frameworks can seamlessly support larger datasets, as shown in [4,5].
>
>
>
>
>
> > Q3. Considering that this research direction ...
>
> A3. We believe that observations about a specific group of models may be overturned, but the core principles will remain unchanged.
> Firstly, our observations about the models are primarily based on existing ones. Therefore, for instance, observation $1$ could be challenged in the future if GFMs capable of outperforming GCN trained from scratch emerge.
> Secondly, our observations regarding the principles will not be overturned. For example, the superiority of GFMs relies on the presence of transferable structural patterns and the incorporation of proper inductive biases into the model. Our principal understanding holds implications for the future design of GFMs. For instance, it's necessary to introduce higher-order structural information to design a GFM that is effective for link prediction tasks, possibly through methods like structural encoding. However, higher structural expressiveness might introduce noise in tasks like node classification, potentially necessitating architectures that disentangle feature and structure modeling.
> We will add a section to discuss the promising future direction based on the conclusion of this benchmark.
>
> [1] Mao, Haitao, et al. "Graph foundation models." arXiv preprint arXiv:2402.02216 (2024).
>
> [2] Hou, Zhenyu, et al. "GraphAlign: Pretraining One Graph Neural Network on Multiple Graphs via Feature Alignment." arXiv preprint arXiv:2406.02953 (2024).
>
> [3] Rossi, Ryan, and Nesreen Ahmed. "The network data repository with interactive graph analytics and visualization." Proceedings of the AAAI conference on artificial intelligence. Vol. 29. No. 1. 2015.
>
> [4] Feng, Jiarui, et al. "TAGLAS: An atlas of text-attributed graph datasets in the era of large graph and language models." arXiv preprint arXiv:2406.14683 (2024).
>
> [5] Hu, Weihua, et al. "Ogb-lsc: A large-scale challenge for machine learning on graphs." arXiv preprint arXiv:2103.09430 (2021).

---

> ### Author Response · Authors · 2024-08-27
>
> Dear reviewers,
>
> We sincerely appreciate your efforts in reviewing this paper and your valuable feedback. We hope our responses have addressed your concerns. If you have any additional questions, please let us know. We look forward to hearing from you.

---

> ### Author Response · Authors · 2024-08-28
> **A friendly reminder**
>
> Dear reviewer 9PU3,
>
> Thanks for your constructive reviews and efforts. As we approach the end of the discussion period, we would greatly appreciate your feedback. We hope that the responses have addressed your concerns. If there are any further questions, please let us know. We look forward to hearing from you.

---

### Official Review · Reviewer_aFAh · 2024-07-24
**A benchmark study for 'modern' cross-task&dataset text-attributed graph learning**

**Rating:** 7
**Confidence:** 3
**Clarity:** The paper is clearly written and easy…

**Review:**

The concept of Text-space Graph Foundation Models presented in this paper is new. It seems like the first benchmark study in this domain, and thus presents challenges in evaluating its contributions as a dataset&benchmark paper. Here are some of my specific points for consideration:

1. Regarding the text-space datasets: The authors claim to introduce novel text-space datasets. However, Section 3 and Figures 2-3 suggest that these datasets are primarily augmentations and recompilations of existing graph datasets and benchmarks. The authors appear to rely on previous works, using templates and LLMs to generate text descriptions for node attributes and labels. This raises two important questions:
(1.a) Could the authors explain their rationale for choosing graph datasets without text attributes instead of those that already contain text attributes, such as WikiKG90Mv2 in OGB?
(1.b) What methods or criteria did the authors use to ensure the quality and accuracy of the generated text descriptions?

2.  Regarding the experiments and insights for co-training over the same task, the experimental results presented in Table 1 raise some questions about the interpretation and scope of the study:
(2.a) The average scores for all methods appear to be very close, with GCN showing the best performance. This seems to contradict Observation 1, which states that "GFM methods present a performance gap compared to GCN training from scratch." Could the authors provide more detailed analysis or statistical tests to support this observation?
(2.b) The study includes only two GFMs (OFA and LLaGA) in this part of the experiment, which seems limited for a comprehensive benchmark study. This limitation is also present in other experiments. Could the authors explain the reasoning behind this choice? Are there constraints in the availability or suitability of other GFMs for such a benchmarking study?

3. Regarding co-training across tasks: Table 6 appears to show results for the OFA model. However, presenting results for only one model in a benchmarking study limits the comparative analysis. Could the authors explain why they chose to focus solely on OFA for this experiment?

4. For Transferring across datasets on the same tasks: The title of this section might be more accurately described as "Transferring across different datasets on the same tasks." The results presented in this section seem logical and align with expectations for transfer learning scenarios.

**Strengths:**

Comprehensive Evaluation Framework: The study incorporates a wide range of settings, including co-training within the same task (node classification, link prediction, and graph classification) as well as across different tasks. It also explores transfer learning both within the same tasks and across various tasks.

Significant Experimental Insights: The paper presents non-trivial observations through extensive experiments. These findings provide valuable insights into the capabilities and limitations of current GFMs, particularly regarding transfer learning, in-context learning, and performance across different graph tasks.

**Additional Feedback:**

None.

**Correctness:**

The evaluation methods and experiments are designed appropriately and performed correctly.

**Documentation:**

The GitHub repository is well documented. Combined with the Appendix, there is sufficient detail to support reproducibility.

**Ethics:**

None.

**Limitations:**

The authors adequately addressed the limitations and potential negative societal impact of their work.

**Opportunities For Improvement:**

A significant limitation of this study is the limited number of text-based GFMs examined. The focus on specific models, like OFA and LLaGA, narrows the generalizability of the findings and may not fully represent the diverse landscape of text-based graph learning approaches. A more accurate title for the current work might be "A Comprehensive Investigation of OFA and LLaGA" instead of "A Comprehensive Benchmark of Text-space GFAs".

**Relation To Prior Work:**

Yes.

**Summary And Contributions:**

The paper investigates Text-space Graph Foundation Models (GFMs), which seek to unify various graph datasets and tasks across multiple domains by utilizing natural language as a shared feature space. It highlights the challenges of creating effective GFMs, particularly the variability in node features among different graphs and the absence of standardized benchmarks. To tackle these issues, the paper introduces a new benchmark for developing text-space GFMs, featuring enhanced datasets, thorough evaluations across diverse scenarios, and fresh insights. The findings indicate that GFM models can identify transferable patterns across different graphs; however, they also face challenges such as negative transfers and the influence of inductive biases in certain situations.

---

> ### Author Rebuttal · Authors · 2024-08-17
>
> Thanks for your valuable feedback.
>
> > Q1. Regarding the text-space datasets: The authors claim ...
>
> A1. We will answer your questions about the novelty of datasets, the selection of datasets, and data quality control.
>
> 1. First, we acknowledge that there might be some confusion in the original statement (line 56 "novel text-space datasets" and Section 3). The primary goal of our paper is not to introduce new datasets for training Graph Foundation Models (GFMs) but rather to transform and leverage qualified datasets to benchmark existing GFMs and explore the potential of unifying feature spaces through text space. We will revise Section 1 and Section 3 accordingly to remove the word "novel".
> 2. Second, we select graphs without native text features (including molecular graphs and graphs using tabular features, such as Tolokers) to evaluate whether text-space GFMs can extend to non-text-attributed graphs. Considering the limited availability of text-attributed graphs, this is crucial for the applicability of text-space GFMs. We don't use knowledge graphs (KG) like WikiKG90Mv2 because the existing GFM frameworks based on subgraphs and graph prompts are unsuitable for addressing KG problems. Firstly, the distinction of KGs lies in their feature space being composed of a finite set of entities, leading to differences in feature encoding [1]. Secondly, although OneForAll also considers KGs, its problem formulation and evaluation differ greatly from traditional KG methods[1]. OneForAll treats KGC as a binary classification problem (using AUC metric), whereas traditional KGC is typically modeled as a retrieval problem (using metrics like MRR). Thirdly, existing subgraph-based GFMs face significant efficiency issues when tackling KG problems. We also acknowledge that treating KGs as text-attributed graphs and using them to augment the training set is an interesting idea, and we have also experimented with this in the following experiment (only OneForAll supports this setting). Here, we co-train together with WN18RR and FB15k237.
>
> |                   | Cora |  CiteSeer   | Pubmed | Average|
> | ----------------- | ---- | --- | -------- | -----|
> | OneForAll(TAG)    | 77.37|  78.52   | 69.43     | 75.11|
> | OneForAll(TAG+KG) | 81.15  | 87.81     | 62.86      | 77.27 |
>
> Preliminary results indicate that co-training with KG has the potential to improve model performance on specific datasets (possibly those sharing similarities with KG). At the same time, it might also negatively impact performance on other datasets.
>
> 3. Third, we follow [2,3] to generate the text descriptions as node, edge, and label attributes. The quality can be ensured from the following aspects: (1) Checking the performance shown in the original paper, for example, the descriptions of molecular properties can achieve good performance in [3]; (2) Comparing the performance of models trained with text-space attributes to the original attributes, with experiments shown in the original paper and Appendix B. In multiple datasets (even for tabular features), we find that the generated attributes get performance closer or even better than the original attributes; (3) Through manual inspection, since most of the generated content primarily consists of label descriptions, the quantity is not very large. We perform manual confirmation after generation. Examples of generated descriptions are shown below. We also find that the generated descriptions have little impact on performance as long as the category name (like "Theory") is included. This may be attributed to the powerful language model used for text encoding, which possesses abundant prior knowledge.
> ```
> Dataset: Cora
> Category: Theory
> Descriptions: The "Theory" category likely refers to research papers that delve into the theoretical aspects of ...
> ```

---

> > ### Author Rebuttal · Authors · 2024-08-17
> >
> > > Q2. Regarding the experiments ...
> >
> > A2. For question 2.a, our observation is that for node classification, if using the same LLM embedding, the existing GFM architectures cannot outperform GCN trained from scratch after co-training. This seems to contradict the conclusion in [2] because the GCN baseline in [2] uses the original TFIDF-like embeddings. We explain this phenomenon in the "further probing" after "observation 1". In summary, we believe the reasons are as follows:
> > 1. **Main reason**: Replacing the original GNN in OFA with a linear GNN improves the performance of GFMs in common homophilous graphs, which indicates the graph structure in these tasks only plays a role similar to feature propagation. This differs from link prediction and graph-level tasks, which require higher-order graph structure patterns, where we observe a clear benefit from co-training.
> > 2. While LLaGA and OneForAll perform well on large-scale datasets with high training ratios, such as Arxiv, they underperform GCN on small datasets with low training ratios, such as Cora. Therefore, their overall average performance is not as good as GCN's. A similar conclusion has been observed in [4] (which focuses on single-dataset evaluation).
> > 3. Co-training requires training with the same hyperparameters, whereas training from scratch allows for hyperparameter tuning based on the dataset.
> >
> > We also conduct statistical tests to verify the results. Here, we assume GFM conducts cross-domain training (average performance is shown).
> >
> > |  | Seed 0 | Seed 1 | Seed 2 |
> > |---|---|---|---|
> > | GCN (single task) | 79.47 | 79.18 | 79.45 |
> > | OneForAll | 78.24 | 78.45 | 78.26 |
> > | GraphMAE | 77.04 | 77.28 | 77.19 |
> > | DGI | 76.96 | 76.85 | 77.02 |
> > | LLaGA | 78.01 | 78.10 | 78.34 |
> >
> > The variance will become small after averaging.
> > Using the ANOVA test, we find that the p-value<0.0005, which means that GCN significantly outperforms other GFM methods.
> >
> > For question 2.b, unlike in natural language processing where LLMs are widely accepted as the foundation models, the graph domain lacks a consensus on the ideal implementation path for GFMs. Therefore, within the unified feature space of text space, we explore potential paths including 1. Foundational graph prompts, as exemplified by OneForAll; 2. GraphLLMs, represented by LLaGA; 3. Self-supervised learning, with GraphMAE as an example; 4. GFMs designed for single tasks, like BUDDY. Our study evaluates all four of these distinct approaches in their respective suitable settings. We have included all general-domain GFMs shown in Page 19 of [5]. Additionally, these two methods (OneForAll and LLaGA) represent a fusion of other methods. OneForAll can be viewed as a simple extension of GNN with text-space features and label embeddings. LLaGA excels in performance and efficiency among GraphLLM architectures (Page 6 of [6]) while maintaining a simple architecture. These characteristics make them the most suitable models for our investigation.
> >
> > > Q3. Regarding co-training across tasks: Table 6 appears to ...
> >
> > A3.  Similar to Problem 2, the reason here is that as far as we know, OneForAll is currently the only model in the text space that simultaneously supports "node-graph", "link-graph", and "node-link-graph" tasks. GCC [1] and GraphMAE [2] also have results at different levels in their original papers, but 1. GCC doesn't support training on graphs with features. 2. GraphMAE formalizes molecular modeling with a masked feature reconstruction in the element space. We have tried using GraphMAE in the text space but found that models can not be well-trained with the direct feature reconstruction objective. Therefore, OneForAll is currently the only model suitable for this scenario.
> > LLaGA is also an available candidate for node and link co-training, and we have included the results in Appendix G.1.1.
> >
> >
> > > Q4. For Transferring across datasets on the same tasks: ....
> >
> > A4. We acknowledge that your proposed title better aligns with the content of this section and will make corresponding changes. Additionally, we will add discussions of important works in graph transfer learning (such as [9]), graph pretraining, and fine-tuning (such as [10,11]) to the related work section.

---

> > > ### Author Rebuttal · Authors · 2024-08-17
> > >
> > > > Q5. A significant limitation of this study is the limited number of ...
> > >
> > > A5. As discussed in question 2, the number of GFMs capable of handling diverse scenarios is limited. We consider four main paths for building GFMs and aim to evaluate these four groups of methods rather than specific baselines. Additionally, we have included experiments with more baselines, as shown in the general response.
> > >
> > >
> > > [1] Zhu, Zhaocheng, et al. "Neural bellman-ford networks: A general graph neural network framework for link prediction." Advances in Neural Information Processing Systems 34 (2021): 29476-29490.
> > >
> > > [2] Liu, Hao, et al. "One for all: Towards training one graph model for all classification tasks." arXiv preprint arXiv:2310.00149 (2023).
> > >
> > > [3] Zhao, Haiteng, et al. "Gimlet: A unified graph-text model for instruction-based molecule zero-shot learning." Advances in Neural Information Processing Systems 36 (2023): 5850-5887.
> > >
> > > [4] Li, Yuhan, et al. "GLBench: A Comprehensive Benchmark for Graph with Large Language Models." arXiv preprint arXiv:2407.07457 (2024).
> > >
> > > [5] Mao, Haitao, et al. "Graph foundation models." arXiv preprint arXiv:2402.02216 (2024).
> > >
> > > [6] Chen, Runjin, et al. "Llaga: Large language and graph assistant." arXiv preprint arXiv:2402.08170 (2024).
> > >
> > > [7]  Qiu, Jiezhong, et al. "Gcc: Graph contrastive coding for graph neural network pretraining." Proceedings of the 26th ACM SIGKDD international conference on knowledge discovery & data mining. 2020.
> > >
> > > [8] Hou, Zhenyu, et al. "Graphmae: Self-supervised masked graph autoencoders." Proceedings of the 28th ACM SIGKDD Conference on Knowledge Discovery and Data Mining. 2022.
> > >
> > > [9] Zhu, Qi, et al. "Transfer learning of graph neural networks with ego-graph information maximization." Advances in Neural Information Processing Systems 34 (2021): 1766-1779.
> > >
> > > [10] Xu, Jiarong, et al. "Better with less: A data-active perspective on pretraining graph neural networks." Advances in Neural Information Processing Systems 36 (2023): 56946-56978.
> > >
> > > [11] Sun, Yifei, et al. "Fine-Tuning Graph Neural Networks by Preserving Graph Generative Patterns." Proceedings of the AAAI Conference on Artificial Intelligence. Vol. 38. No. 8. 2024.

---

> > > > ### Comment · Reviewer_aFAh · 2024-08-23
> > > > **Response from Reviewer aFAh**
> > > >
> > > > > Preliminary results indicate that co-training with KG has the potential to improve model performance on specific datasets (possibly those sharing similarities with KG). At the same time, it might also negatively impact performance on other datasets.
> > > >
> > > > The results are very significant here, but I'm personally not sure whether the explanation is convincing—is the overlap of KG dataset and Citeseer greater than the other two datasets? A more detailed analysis of the relationship between KG co-training and dataset characteristics might provide deeper insights.
> > > >
> > > > > Third, we follow [2,3] to generate the text descriptions as node, edge, and label attributes. The quality can be ensured from the following aspects: ...
> > > >
> > > > Though you provide justification, I'm still not convinced [2,3] is the best way (or even a good way) for dataset conversion. As we all know, different prompt designs will deeply affect LLM's performance. As this paper mainly focuses on benchmarking GFMs under different transfer learning settings, I think it can be left for future work to explore optimal prompt designs for dataset conversion.
> > > >
> > > > > A2. For question 2.a, our observation is that for node classification, ....
> > > >
> > > > The results and analysis here are useful, especially for newcomers transitioning from GNN to GFM field. I stronglyrecommend adding this part to the manuscript.
> > > >
> > > > ---
> > > > I appreciate the authors' effort in conducting these experiments. I believe it will be a solid benchmark study after these revisions. As nowadays the definition of GFM is vague and rapidly changing, the new taxonomy of GFM introduced in this paper, although not optimal, can bring new insights to this community.
> > > >
> > > > Therefore, I support the acceptance of this paper,

---

> > > > > ### Author Rebuttal · Authors · 2024-08-23
> > > > >
> > > > > First, we would like to thank you for your feedback and support.
> > > > >
> > > > > > Q1. The results are very significant here, but I'm personally not sure whether...
> > > > >
> > > > > Although we guarantee the authenticity of our results, we also acknowledge that the results on these small datasets are not sufficient to draw a comprehensive conclusion. One factor limiting us from conducting larger-scale experiments is the model's efficiency. For instance, OneForAll already requires nearly 20 hours of training on this data (30 epochs). One further possible explanation for the results is that using only MLP on Cora-TAG and Citeseer-TAG can achieve results even better than OneForAll, indicating that semantic information is useful on these two datasets, and the addition of KG may compensate in this aspect.
> > > > > | Single (not co-train) | Cora | CiteSeer | Pubmed |
> > > > > |---|---|---|---|
> > > > > | MLP | 83.22 | 91.95 | 49.88 |
> > > > > | OneForAll | 77.37 | 78.52 | 69.43 |
> > > > > | BUDDY | 91.37 | 96.57 | 83.29 |
> > > > >
> > > > > We acknowledge that the conclusions drawn through OneForAll are not very representative (but unfortunately, we don't have better choices right now) because it does not design effective structural embedding. We believe that future work can be carried out from the following two perspectives:
> > > > >
> > > > > 1. Continue the exploration of text-space GFM and design an efficient (linear) GNN that can effectively utilize edge embedding (the semantics of edges on KG are very important), and can scale up to large datasets. Then, we study large-scale co-training based on this model.
> > > > > 2. Following the idea of GNN for KG, explore how to design a tokenizer [1, 2] to unify generalized text-attributed graphs with KG, and then explore the effects after co-training.
> > > > >
> > > > >
> > > > > > Q2. Though you provide justification, ...
> > > > >
> > > > > You present a very valuable future direction.
> > > > > Currently, the overall transformation approach into text space is as follows:
> > > > > 1. Convert information from different modalities into text. For example, images can be converted using CLIP[3], and tabular features can be converted directly by writing text prompts[4].
> > > > > 2. Convert the transformed text into text embeddings using a unified text encoder.
> > > > >
> > > > > In both steps 1 and 2, there is manual prompt design involved. We admit that we have not conducted in-depth ablation studies in this aspect but mainly follow previous works, and we have not yet come up with a more promising solution. A possible future direction is to use some automatic optimization methods [5] to get rid of handcrafted engineering.
> > > > >
> > > > >
> > > > > > Q3. The results and analysis here are useful, especially for newcomers transitioning from GNN to GFM field. I stronglyrecommend adding this part to the manuscript.
> > > > >
> > > > > We briefly discuss this phenomenon in lines 237 to 251. Due to space constraints, we abbreviate some parts, which may have caused confusion for readers. In the revision, we will supplement the original discussion based on the rebuttal's content.
> > > > >
> > > > >
> > > > > [1] Rajput, Shashank, et al. "Recommender systems with generative retrieval." Advances in Neural Information Processing Systems 36 (2024).
> > > > >
> > > > > [2] Yang, Ling, et al. "VQGraph: Rethinking Graph Representation Space for Bridging GNNs and MLPs." arXiv preprint arXiv:2308.02117 (2023).
> > > > >
> > > > > [3] Radford, Alec, et al. "Learning transferable visual models from natural language supervision." International conference on machine learning. PMLR, 2021.
> > > > >
> > > > > [4] Ye, Chao, et al. "CT-BERT: learning better tabular representations through cross-table pre-training." arXiv preprint arXiv:2307.04308 (2023).
> > > > >
> > > > > [5] Yuksekgonul, Mert, et al. "TextGrad: Automatic" Differentiation" via Text." arXiv preprint arXiv:2406.07496 (2024).

---

### Official Review · Reviewer_Ltcu · 2024-07-25

**Rating:** 7
**Confidence:** 3

**Review:**

Please see the comments in the “Strengths” and “Opportunities For Improvement” sections below.

**Strengths:**

**S1. The proposed benchmark provides multiple text-space datasets, GFM models, and evaluation settings.**
This work improves upon earlier works that involve text-space GFMs by including a larger number of text-space datasets drawn from several domains, different types of GFM models like graph SSL and foundational graph prompt methods, and multiple evaluation settings that cover four GFM training paradigms as well as node-, link- and graph-level tasks. With all these components combined, the proposed benchmark provides a more comprehensive environment for text-space GFM evaluation than existing works.

**S2. This work performs a systematic investigation of text-space GFMs’ performance in various scenarios.**
Based on the multiple application scenarios that the proposed benchmark supports, this work conducts extensive experiments, demonstrating the effectiveness as well as the weaknesses of existing text-space GFMs. These experimental results can inform future research directions, e.g., investigating some of the scenarios existing text-space GFMs cannot handle successfully, and developing more effective solutions.

**Additional Feedback:**

I have no additional feedback.

**Clarity:**

The paper is overall well organized, while it is sometimes hard to follow as discussed above.

**Correctness:**

This paper presents both datasets and a benchmark. Dataset construction and experiments are performed correctly to my knowledge.

**Documentation:**

The paper presents sufficient details about the datasets, including the references for the original data sources, their licenses, as well as a URL for the dataset and plans for hosting and maintenance. Also, the paper provides details of experimental setup, e.g., hyperparameter settings of the methods used for evaluation.

**Ethics:**

No. I believe there are no ethical concerns with this paper that warrant further discussion or review.

**Limitations:**

Yes, authors discuss some of the limitations and potential negative impact of their work in the Appendix.

**Opportunities For Improvement:**

**I1. The novelty and contributions of the proposed datasets is limited.**
While the development of text-space datasets is one of the major contributions of this work, the proposed datasets seem to be mostly a collection of existing graphs, which were developed and used by prior studies. Although the transformation of node attributes into texts requires some additional processing to construct text-space graph data, such transformations also utilize approaches proposed by previous works, limiting the novelty and contribution of the proposed datasets.

**I2. Writing can be improved.**
While the paper is overall well organized, it is sometimes hard to follow as it assumes familiarity with some of the existing works. For instance, it is not clear how this work transforms the node attributes into texts, as the paper only says that the transformation process follows existing works; similarly, the meaning of neural scaling properties is not clear.
Also, the paper discusses results for the pre-training scenario much more briefly than the co-training setting, which makes it hard to follow without being familiar with the evaluation settings (e.g., how graph in-context learning is done) and the methods used for this scenario (e.g., why conducting no pre-training is preferable as to the use of simple SBERT). Providing some more details and discussions could make the paper more accessible and self-contained.

**Relation To Prior Work:**

The paper discusses several prior works involving text-space graph foundation models, and how this work addresses some of their limitations.

**Summary And Contributions:**

Graph foundation models (GFMs) aim to handle different graphs and tasks with a unified backbone. To address the challenge that different graphs often exhibit diverse node features, text-space GFMs transform node features into text and apply large language models to generate embeddings for different graphs in a unified feature space.
Despite the potential of text-space GFMs, there are two major limitations that hinder a comprehensive understanding of their effectiveness in different application settings, which are the absence of a comprehensive benchmark, and the limited number of datasets used for evaluation. This paper aims to tackle these two problems by developing a benchmark that provides text-space datasets and comprehensive evaluation settings. Then based on the proposed benchmark, this work conducts extensive experiments and discusses the results.

---

> ### Author Rebuttal · Authors · 2024-08-17
>
> Thanks for your valuable feedback.
> > Q1. The novelty and contributions of the proposed ...
>
> A1: We acknowledge that our original statement may lead to confusion (line 56, "novel text-space datasets" and Section 3). The primary goal of our paper is not to collect brand new datasets for training Graph Foundation Models (GFMs) but rather to transform and leverage qualified datasets to benchmark existing text-space GFMs. We will revise Section 1 and Section 3 accordingly to remove the word "novel."
>
> We choose to transform existing datasets for the following reasons: 1. Since we are studying text-space datasets, the attributes of these datasets themselves need to be convertible to text in some way. We have considered most datasets satisfying this condition, and many large-scale datasets without valid features like [1] are unsuitable; 2. These are verified high-quality datasets, making our results more reliable and generalizable. During data collection, we also filter out some low-quality data (like DBLP), such as those where the performance of MLP and GCN are almost the same. Moreover, in addition to commonly used datasets like Cora, Citeseer, and OGB, we also introduce new text-space datasets such as Amazon rating and Tolokers. Our proposed data representation framework can also effectively support new relevant datasets, such as [2].
>
>
> > Q2. Writing can be improved. While the paper is ...
>
> A2. We acknowledge the three omissions in our writing that you point out: 1. There are no details regarding the dataset preprocessing, specifically the generation of text descriptions; 2. There is no discussion of neural scaling properties; and 3. There is no introduction to the pretraining baseline. We will provide a brief overview here and outline our future revision plan.
> 1. For attribute preprocessing, we follow [3,4] to conduct preprocessing. For example, given an E-commerce dataset, we set the node attribute to "Feature node. Product Title: <product_title>" (<product_title> is the text-space attribute we collect), edge attribute to "Feature edge. These two items are frequently co-purchased or co-viewed". Label description is generated by LLMs, with examples like "The "Case Based" category refers to research papers focusing on case-based reasoning (CBR) in the field of artificial intelligence...". We double-check the outputs of LLMs to ensure the quality of their generation. Then, we encode these attributes into corresponding feature, edge, and label embedding using a text encoder model (such as SentenceBERT).
> 2. Neural scaling [5] is a phenomenon where the capabilities of a model improve as the number of model parameters and the amount of data increase. We primarily focus on data-level neural scaling. In past GNN training, the focus is mainly on a single dataset. So, taking node-level tasks as an example, we observe the neural scaling with the number of nodes or edges within the same dataset [6]. We can examine neural scaling across different datasets after adopting text-space features to unify the feature space. In this paper, we check the effects of scaling by comparing models trained from scratch, datasets from similar domains, and datasets from diverse domains. We observe the benefits when the following conditions are met: 1. Transferable structural patterns exist, and 2. The model possesses the inductive bias required for the task.
> 3. Graph in-context learning is typically achieved through the foundational graph prompt in section 2.2. Here, we take OneForAll[1] as an example. It first introduces prompt nodes by converting original labels into augmented graph nodes and connecting them to the ego-graph. Then, it unifies tasks at different levels by viewing node classification as ego-graph classification and link prediction as the classification of the node pair-induced subgraph. For few-shot in-context learning, the subgraphs of labeled nodes are extracted and connected to the prompt nodes with identical labels. For "no pretrain" in SBERT, this experiment aims to find the critical component for graph in-context learning. So, we directly use the propagated SBERT embeddings for prediction and find that the results are close to or surpass most graph in-context learning baselines. With this experiment, we aim to demonstrate that the performance of current graph in-context learning methods mainly stems from LLM embeddings, while the gains from graph pretraining are pretty limited. The exception is observed with Amazon ratings (a heterophilous dataset), where pretraining provides a distinct advantage.
>
> **Revision plan.** We will do the following revisions
> * Adding one paragraph in Section 3 to detail the preprocessing process.
> * Adding explanations on neural scaling properties in Section 2.1.
> * Revise Section 2.2 to introduce background on how graph in-context learning is implemented with foundational graph prompts. Revise 4.4 to give more background on simple SBERT baselines and the objective of this experiment.
>
> [1] Rossi, Ryan, and Nesreen Ahmed. "The network data repository with interactive graph analytics and visualization." Proceedings of the AAAI conference on artificial intelligence. Vol. 29. No. 1. 2015.
>
> [2] Feng, Jiarui, et al. "TAGLAS: An atlas of text-attributed graph datasets in the era of large graph and language models." arXiv preprint arXiv:2406.14683 (2024).
>
> [3] Liu, Hao, et al. "One for all: Towards training one graph model for all classification tasks." arXiv preprint arXiv:2310.00149 (2023).
>
> [4] Zhao, Haiteng, et al. "Gimlet: A unified graph-text model for instruction-based molecule zero-shot learning." Advances in Neural Information Processing Systems 36 (2023): 5850-5887.
>
> [5] Kaplan, Jared, et al. "Scaling laws for neural language models." arXiv preprint arXiv:2001.08361 (2020).
>
> [6] Liu, Jingzhe, et al. "Neural scaling laws on graphs." arXiv preprint arXiv:2402.02054 (2024).

---

> > ### Comment · Reviewer_Ltcu · 2024-08-23
> >
> > Thank you for your response. It has addressed some of my concerns and questions. At the same time, note that what I mentioned in my review in I2 are some of the examples that can be improved. It is recommended to make revisions and clarifications in other places as necessary to make the paper more self-contained and accessible, as this work builds upon and discusses several recent works in a rapidly evolving area. The rebuttal also presented additional experimental results using several new baselines as well as discussion of new results, which further improves the paper. In light of these changes, I increased my score.

---

> > > ### Author Rebuttal · Authors · 2024-08-23
> > >
> > > Thank you for your valuable feedback and support. We acknowledge that the writing should be further improved, and your suggestions are valuable in refining the quality of this paper. We will make corresponding changes based on your feedback in the revision.

---

### Author Rebuttal · Authors · 2024-08-17

Thanks for all the valuable feedback. We are willing to clarify some common points raised by reviewers.

* **Scope and novelty of the benchmark**: We acknowledge our original statement may lead to confusion (line 56 "novel text-space datasets" and Section 3) that our benchmark aims at collecting new graphs for training GFMs. We want to clarify that our primary goal is to transform existing datasets into text space and study the potential of developing GFMs in this space. Towards this goal, we evaluate four roads to develop GFMs: 1. Foundational graph prompts, as exemplified by OneForAll; 2. GraphLLMs, represented by LLaGA; 3. Self-supervised learning, with GraphMAE as an example; 4. GFMs designed for single tasks, like BUDDY. Our study evaluates all four of these distinct approaches in their respective suitable settings. Based on our experimental results, we come up with valuable inspirations for next-step development.
* **Focus on co-training**: The original manuscript's background introduction and emphasis on the transferring setting are insufficient. In the revision, we will address this by 1. Adding a more comprehensive introduction to the transferring problem setting, and 2. Incorporating additional baselines like GraphLLM, Graphtext, GraphAdapter, and GPPT. We focus on co-training settings for the following reasons: 1. GFM development is still in its early stages, and our main goal is to understand the potential unifying features in the text space to develop GFMs. We aim to derive principles from experimental results to guide future model design. In this sense, co-training better suits our needs since they can directly reflect the model's performance while transferring also involves adapting pre-trained models to downstream tasks, which is a further step based on effective co-training. 2. Co-training is an essential ingredient for efficient transferring [1]. Considering the limited size of a single graph, it's necessary to co-train them together to scale models.
* **More experimental results**:
1. Effects of introducing KG into co-training: co-training with KG can improve model performance on specific datasets (possibly those sharing similarities with KG). At the same time, it might also negatively impact performance on other datasets.

|                   | Cora |  CiteSeer   | Pubmed | Average|
| ----------------- | ---- | --- | -------- | -----|
| OneForAll(TAG)    | 77.37|  78.52   | 69.43     | 77.27|
| OneForAll(TAG+KG) | 81.15  | 87.81     | 62.86      | 74.05 |

2. Statistical test: The variance will become small after averaging.
Using the ANOVA test, we find that the p-value<0.0005, which means that GCN significantly outperforms other GFM methods.

|  | Seed 0 | Seed 1 | Seed 2 |
|---|---|---|---|
| GCN (single task) | 79.47 | 79.18 | 79.45 |
| OneForAll | 78.24 | 78.45 | 78.26 |
| GraphMAE | 77.04 | 77.28 | 77.19 |
| DGI | 76.96 | 76.85 | 77.02 |
| LLaGA | 78.01 | 78.10 | 78.34 |

3. More baselines for transferring setting and more graph-llms: For few-shot inference, two strategies are shown effective: 1. LLM embedding with proper graph inductive bias, and then the task should be reformulated as finding the closet class labels based on similarity; 2. Pure LLM with text inputs (projector doesn't work), and the downstream task should be correlated to the node feature semantics (for ratings, it's a rating prediction task).
* Semantic label embeddings are critical to the transferability of graph prompts. Otherwise, they fail like GPPT and GPrompt.

|  | Cora |  |  | History |  |  | Ratings |  |  |
|:---:|:---:|:---:|:---:|:---:|:---:|:---:|:---:|:---:|:---:|
|  | 0 shot | 3 shot | FT | 0 shot | 3 shot | FT | 0 shot | 3 shot | FT |
| LLaGA | 18.25 | 60.7 | 80.45 | 22.05 | 36.45 | 82.55 | 23.15 | 23.45 | 28.2 |
| Prodigy | N/A | 71.60 | N/A | N/A | 52.66 | N/A | N/A | 67.41 | N/A |
| Simple SBERT | 67.41 | 68.42 | 82.2 | 59.25 | 51.25 | 85.3 | 27.39 | 20.95 | 48.46 |
| GraphText | N/A | 50.33 | N/A | N/A | 48.00 | N/A | N/A | 37.67 | N/A |
| GraphAdapter | 21.09 | 33.74 | 62.56 | 17.43 | 36.60 | 82.94 | 27.64  | 29.41 | 38.69 |
| GraphLLM | 67.33 | 68.00 | N/A | 34.67 | 56.67 | N/A | 24.33 | 34.33 | N/A |
| GraphGPT | 22.50 | N/A | N/A | 15.50 | N/A | N/A | 24.00 | N/A | N/A |
| GPPT | N/A | 44.14 |	65.84 | N/A | 27.54 | 35.94 | N/A | 14.24 | 20.22 |
| GPrompt | N/A | 55.38 | 70.82 | N/A |17.36 |21.33 | N/A |15.38 |17.23 |

---

> ### Author Rebuttal · Authors · 2024-08-17
>
> 4. More baselines for graph classification and prompt-based methods
>
> We apply GraphCL, GPPT, GPrompt, and GPF to the graph classification task. We compare the following two ways to use GraphCL: 1. Pre-training it on ZINC with original chemical features and then fine-tuning; 2. Training and fine-tuning it on downstream task data. For GPPT and Gprompt, we also consider pre-training on ZINC. For methods pre-trained on ZINC, the performance on PCBA is not available, considering the duplication between PCBA and ZINC.
>
> |  | pcba | hiv | tox21 | bace | bbbp | muv | toxcast |
> |---|---|---|---|---|---|---|---|
> | OneForAll | 23.61 | 75.24 | 82.5 | 77.32 | 69.97 | 70.39 | 68.39 |
> | GraphCL (co-train) | 20.26 | 70.33 | 73.74 | 64.42 | 63.09 | 73.62 | 60.77 |
> | GraphCL (Zinc, transferring) | N/A | 78.47 | 73.87 | 75.38 | 69.68 | 69.8 | 62.4 |
> | EdgePred-GPPT | N/A | 73.19 | 78.91 | 70.31 | 69.43 | 82.06 | 64.86 |
> | EdgePred-GPrompt | N/A | 59.31 | 68.09 | 67.70 | 69.29 | 62.35 | 60.54 |
> | GPF | N/A | 77.60 | 79.74 | 81.57 | 69.57 | 82.86 | 65.65 |
>
> From the experimental results, we have the following observations:
> 1. If we directly compare the co-training setting (pre-train and finetune on the same data set), supervised methods like OneForAll present clear benefits, which can be attributed to unifying the label space.
> 2. GraphSSL and graph prompt methods are effective when pre-training on large-scale external data (like ZINC), and methods like GPF with learnable token embedding show the best performance.
> 3. The trade-off between few-shot and fine-tuning performance is a challenge here. GPrompt reformulates the task formulation as finding closet labels in the embedding space and thus can not utilize labels as well as traditional methods.
>
> We further apply GPPT and GPrompt to the node-level tasks, and compare them with other SSL methods, such as GraphMAE and OneForAll. We co-train models on citation datasets and evaluate them on Cora and CiteSeer.
>
> |  | Cora | CiteSeer |
> |---|---|---|
> | OneForAll | 70.74 | 81.66 |
> | GraphMAE | 78.09 | 68.8 |
> | GPPT | 70.92 | 63.38 |
> | Gprompt | 68.13 | 54.95 |
>
> We find that prompt-based models GPPT and Gprompt are ineffective here. The main reason is that they are designed using numeric labels instead of semantic label embeddings, which doesn't present good transferability across datasets.
>
>
> [1] Li, Yuhan, et al. "ZeroG: Investigating Cross-dataset Zero-shot Transferability in Graphs." arXiv preprint arXiv:2402.11235 (2024).

---

### Author Response · Authors · 2024-08-22

Dear reviewers,

We sincerely appreciate your efforts in reviewing this paper and your valuable feedback. We hope our responses have addressed your concerns. If you have any additional questions, please let us know. We look forward to hearing from you

---

### Author Response · Authors · 2024-08-31

Dear Reviewers,

We want to express our sincere gratitude for your detailed feedback on our paper. As we are nearing the end of the discussion period, we would greatly appreciate your prompt attention to our rebuttal and any subsequent comments. If you have any further questions or need additional clarification, please do not hesitate to reach out.

Thank you,

All Authors

---

### Decision · Program_Chairs · 2024-09-26

**Decision:**

Accept (Poster)

**Comment:**

In this submission, the authors introduce text-space datasets and conduct comprehensive evaluations. Most of the reviewers agree that this work has great potential. Besides, Reviewer SB91 also states that the rebuttal from authors "have addressed most of my concerns, and" it is indeed a good paper but needs to be carefully revised for more readable". Given these, I suggest to accept this submission, and hope the authors can make this submission more strong when preparing the final version.